# Endotyping-informed therapy for patients with chest pain and no obstructive coronary artery disease: a randomized trial

Patients undergoing invasive coronary angiography for the investigation of chest pain commonly do not have obstructive coronary artery disease. In contemporary practice, most of these individuals do not undergo functional diagnostic tests, leaving the cause of the chest pain uncertain. Stress cardiovascular magnetic resonance imaging (MRI) can be used to measure myocardial blood flow, detect coronary microvascular dysfunction and endotype individual patients, but evidence of clinical utility from randomized trials is lacking. This study was a prospective, multicenter, parallel group, 1:1 randomized, controlled superiority trial of adenosine stress cardiovascular MRI-guided management in 250 patients (mean age, 63.3 years; 50.4% female) with chest pain and unobstructed coronary arteries. The primary outcome of the diagnostic study, defined as the reclassification of the initial diagnosis based on the angiogram, occurred in 131 patients (53.0% (95% confidence interval: 46.6−59.3%); $P < 0.001$), indicating that the primary outcome for the diagnostic study was met. The primary outcome of the randomized trial was the Seattle Angina Questionnaire (SAQ) summary score at 12 months after randomization. The mean ± s.d. SAQ summary scores at 12 months in the intervention and control groups were 70.9 ± 23.6 (21.7 ± 22.6 change from baseline) and 52.1 ± 24.1 (−0.8 ± 20.4 change from baseline) (adjusted mean difference: 20.9 (95% confidence interval: 15.8−26.0)), respectively, indicating that the primary outcome of the randomized trial was met. Improvements were also observed in the prespecified secondary outcome of the EQ-5D-5L questionnaire at 12 months (adjusted mean difference 0.09 (95% confidence interval: 0.04−0.13)). In this study of patients with chest pain and unobstructed coronary arteries, endotyping-informed therapy revised the diagnosis in more than half of the participants and improved angina and health-related quality of life. ClinicalTrials.gov identifier: NCT04805814.

✉ e-mail: colin.berry@glasgow.ac.uk

Diagnosing the cause of stable chest pain can be challenging. The causes include angina due to obstructive coronary artery disease, microvascular angina and vasospastic angina (that is, angina with unobstructed coronary arteries (ANOCA)/ischemia with no obstructed coronary arteries (INOCA)) or a non-cardiac etiology. The diagnostic management of chest pain is described in guidelines[1,2]. Based on functional imaging, the prevalence of endotypes of ANOCA in patients referred for diagnostic evaluation is approximately 40%[3] and 43% (range, 16–73%)[4], respectively. Coronary angiography provides an anatomical assessment of atherosclerosis, whereas the evaluation of vasomotor causes of angina requires functional testing[1,2].

Coronary computed tomography angiography (CCTA) is recommended as a first-line test for the evaluation of suspected angina[1,2]. However, in patients selected for invasive coronary angiography with or without prior CCTA, half have no obstructed coronary arteries identified[5]. In contemporary practice, most of these individuals do not undergo functional tests.

Coronary microvascular dysfunction is associated with persistent anginal symptoms[6], impaired quality of life[7], increased risk of major adverse cardiovascular events[8,9] and considerable health resource utilization due to recurrent hospitalizations and repeat coronary angiography[8]. The Coronary Microvascular Angina (CorMicA) trial demonstrated that coronary function testing during invasive management of suspected ANOCA is associated with improvements in symptoms, health-related quality of life and health economics[10]. These results were recently validated in the ILIAS ANOCA trial[11]. However, the effect of adjunctive diagnostic tests differs according to the test modality and population[12]. Cardiovascular magnetic resonance (CMR)[13,14] and nuclear stress positron emission tomography[15] quantify myocardial blood flow and may be used to diagnose coronary microvascular dysfunction, but evidence of clinical utility from randomized trials is lacking.

Among patients with chest pain without obstructive coronary artery disease defined by invasive angiography, we aimed to assess whether re-evaluation of the endotype informed by stress CMR imaging (myocardial blood flow disclosed to inform endotyping) versus no functional assessment (myocardial blood flow measured but not disclosed), with endotype-specific treatment algorithms applied to all participants, affected the final diagnosis, angina burden and health-related quality of life.

We hypothesized that coronary vasomotor endotypes are prevalent in a post-angiography population with no obstructive coronary arteries, and, compared to angiography-guided management, CMR-guided management changes the diagnosis and improves symptoms and health-related quality of life.

## Results

Between 9 February 2021 and 18 August 2023, 273 outpatients who had undergone clinically indicated invasive coronary angiography for investigation of chest pain and had a report of no obstructive coronary artery disease were screened and provided written informed consent within 3 months of the angiogram (Extended Data Fig. 1).

Two hundred and fifty of these patients attended for CMR imaging and were randomized before the scan (Table 1 and Fig. 1). One randomized participant was unable to complete the scan due to claustrophobia. Twenty-three patients who had provided consent did not attend for CMR imaging due to logistical and other reasons (Fig. 1).

### Randomized population: observations at baseline

Two hundred and fifty participants (91.6%) were randomized (*n* = 126 control group, *n* = 124 intervention group) (Table 1): mean age was 63.3 years; 50.4% were female; and 17.2% had diabetes mellitus. One hundred and five (42.0%) participants had a history of hospitalization for chest pain; 56 (22.4%) had previously undergone coronary angiography;

41 (16.4%) had a history of previous percutaneous coronary intervention; and 30 (12.0%) had a history of prior myocardial infarction.

Most patients (69.2%) had undergone non-invasive diagnostic stress testing before invasive coronary angiography, and most (87.3%) of these participants had undergone treadmill exercise tolerance testing. A minority (15.6%) of participants had undergone CCTA imaging (Table 1). Medicines for the prevention and treatment of angina were commonly prescribed (Table 1).

The findings from invasive coronary angiography are described in Supplementary Table 1. Following standard care invasive coronary angiography prior to enrollment, 244 (97.6%) of the participants had been diagnosed by the attending cardiologist as having non-cardiac chest pain. A minority (8.4%) of participants had received an adjunctive functional test of coronary artery disease severity or microvascular function during invasive management.

After completion of the SAQ at baseline (Table 1), the mean angina frequency score was 55.8 ± 25.5, corresponding with weekly angina (SAQ frequency score 31–60 indicates weekly angina, 61–99 indicates monthly angina). The mean SAQ angina limitation score was 53.6 ± 25.0, corresponding with mild to moderate angina limitation. Overall, the angina burden of the patient population was mostly consistent with Canadian Cardiovascular Society class II–III angina, with a mean SAQ summary score of 51.1 ± 20.7 (Table 1).

### Non-invasive endotyping and findings after CMR imaging in all participants

The endotypes based on myocardial perfusion imaging are described in Table 2. Two hundred and fifty participants attended for adenosine stress CMR imaging and were randomized, and myocardial blood flow was quantified in 249 (99.6%) of these individuals (Fig. 1). The mean global myocardial blood flow during adenosine stress was 2.3 (0.6) ml min$^{-1}$ g$^{-1}$.

### Diagnostic study

Diagnoses per participant after CMR imaging were 127 (51.0%) microvascular angina, one (0.4%) vasospastic angina, zero (0.0%) obstructive coronary artery disease, 117 (47.0%) non-cardiac chest pain and four (1.6%) other diagnoses (Table 2).

**Primary outcome of the diagnostic study.** A bar chart of diagnoses at sequential timepoints is shown in Extended Data Fig. 2. Of 250 participants, one patient did not have a CMR diagnosis (Fig. 1), and their 'post-CMR' diagnosis was taken as the same as their angiography-guided diagnosis (non-cardiac chest pain).

The primary outcome of the diagnostic study, defined as the reclassification of the initial diagnosis based on the angiogram, occurred in 131 patients (53.0% (95% confidence interval: 46.6–59.3%); *P* < 0.001) (Table 2). Notably, the randomized groups did not differ in their original diagnoses (*P* = 0.683; Table 2) nor in their post-CMR diagnoses (*P* = 0.905; Table 2) but did differ markedly in the final working diagnosis, after disclosure or non-disclosure of non-invasive endotyping results that were used to guide subsequent treatment (Table 3). In the control arm, one patient (0.8%) had a diagnosis of microvascular angina based on medical history and coronary angiogram results. In the intervention arm, after disclosure of non-invasive endotyping results, 62 (50.0%) had a diagnosis of microvascular angina, a difference of 49.2% (95% confidence interval: 41.3–58.7%). A diagnosis of non-cardiac chest pain was predominant in the control arm (98.4%) and much reduced in the intervention arm (48.4%).

### Nested randomized trial

Follow-up continued until 1 September 2024, representing a median (interquartile range) follow-up period of 386 (371.0–398.0) days and 383 (370.0–396.0) days in the intervention and control groups, respectively. Follow-up visits were completed in all participants at 6 months and

**Table 1 | Baseline demographic and clinical characteristics for the randomized population**

| Characteristic | All | Control | Intervention |
|---|---|---|---|
| | *n*=250 | *n*=126 | *n*=124 |
| Demographics | | | |
| Age (years), mean (s.d.) | 63.3 (9.2) | 64.0 (8.9) | 62.5 (9.4) |
| Sex | | | |
| Male, *n* (%) | 124 (49.6%) | 60 (47.6%) | 64 (51.6%) |
| Female, *n* (%) | 126 (50.4%) | 66 (52.4%) | 60 (48.4%) |
| Ethnicity | | | |
| White, *n* (%) | 241 (96.4%) | 121 (96.0%) | 120 (96.8%) |
| Asian, *n* (%) | 8 (3.2%) | 5 (4.0%) | 3 (2.4%) |
| Black, *n* (%) | 1 (0.4%) | 0 (0.0%) | 1 (0.8%) |
| BMI (kg m$^{-2}$), mean (s.d.) | 30.3 (5.1) | 30.2 (5.4) | 30.5 (4.7) |
| BMI ≥ 30 kg m$^{-2}$, *n* (%) | 128 (51.2%) | 60 (47.6%) | 68 (54.8%) |
| Waist circumference (cm), mean (s.d.) | 97.1 (13.4) | 96.2 (14.0) | 98.1 (12.7) |
| Smoking status | | | |
| Current smoker, *n* (%) | 21 (8.4%) | 12 (9.5%) | 9 (7.3%) |
| Former smoker, *n* (%) | 83 (33.2%) | 44 (34.9%) | 39 (31.5%) |
| Never smoked, *n* (%) | 146 (58.4%) | 70 (55.6%) | 76 (61.3%) |
| Previous coronary angiogram, *n* (%) | 56 (22.4%) | 29 (23.0%) | 27 (21.8%) |
| Previous percutaneous coronary intervention, *n* (%) | 41 (16.4%) | 20 (15.9%) | 21 (16.9%) |
| Previous myocardial infarction, *n* (%) | 30 (12.0%) | 17 (13.5%) | 13 (10.5%) |
| Previous stroke or transient ischemic attack, *n* (%) | 11 (4.4%) | 4 (3.2%) | 7 (5.6%) |
| Hospitalization for chest pain, *n* (%) | 105 (42.0%) | 49 (38.9%) | 56 (45.2%) |
| History of valve disease, *n* (%) | 6 (2.4%) | 2 (1.6%) | 4 (3.2%) |
| Hypertension, *n* (%) | 117 (46.8%) | 61 (48.4%) | 56 (45.2%) |
| Diabetes mellitus, *n* (%) | 43 (17.2%) | 22 (17.5%) | 21 (16.9%) |
| Atrial fibrillation, *n* (%) | 19 (7.6%) | 10 (7.9%) | 9 (7.3%) |
| Chronic obstructive pulmonary disease, *n* (%) | 35 (14.0%) | 16 (12.7%) | 19 (15.3%) |
| Systolic blood pressure (mmHg), mean (s.d.) | 138.5 (17.7) | 136.8 (16.2) | 140.1 (19.0) |
| Diastolic blood pressure (mmHg), mean (s.d.) | 77.5 (10.0) | 76.2 (8.7) | 78.7 (11.1) |
| Charlson comorbidity index, mean (s.d.) | 2.4 (1.4) | 2.5 (1.4) | 2.3 (1.4) |
| QRISK3-predicted 10-year risk of myocardial infarction or stroke, mean (s.d.)[a] | 17.2 (12.1) | 18.0 (13.2) | 16.3 (10.9) |
| Preventive therapy | | | |
| Aspirin, *n* (%) | 213 (85.2%) | 103 (81.7%) | 110 (88.7%) |
| Statin, *n* (%) | 223 (89.2%) | 109 (86.5%) | 114 (91.9%) |
| Ezetimibe, *n* (%) | 5 (2.0%) | 1 (0.8%) | 4 (3.2%) |
| Angiotensin-converting enzyme inhibitor, *n* (%) | 75 (30.0%) | 47 (37.3%) | 28 (22.6%) |
| Angiotensin receptor blocker, *n* (%) | 23 (9.2%) | 8 (6.3%) | 15 (12.1%) |
| Angina medication | | | |
| Beta-blocker, *n* (%) | 196 (78.4%) | 95 (75.4%) | 101 (81.5%) |
| Calcium channel blocker, *n* (%) | 78 (31.2%) | 47 (37.3%) | 31 (25.0%) |
| Nitrates, *n* (%) | 119 (47.6%) | 59 (46.8%) | 60 (48.4%) |
| Nicorandil, *n* (%) | 32 (12.8%) | 20 (15.9%) | 12 (9.7%) |
| Glyceryl trinitrate spray, *n* (%) | 199 (100.0%) | 100 (100.0%) | 99 (100.0%) |
| Clinical characteristics | | | |
| Cholesterol and lipid profile | | | |
| Total cholesterol (mmol l$^{-1}$), mean (s.d.) | 4.0 (1.0) | 4.1 (1.1) | 4.0 (1.0) |
| HDL cholesterol (mmol l$^{-1}$), median [Q1, Q3] | 1.3 [1.1, 1.6] | 1.3 [1.1, 1.6] | 1.3 [1.1, 1.5] |

**Table 1 (continued) | Baseline demographic and clinical characteristics for the randomized population**

| Characteristic | All | Control | Intervention |
|---|---|---|---|
| | *n*=250 | *n*=126 | *n*=124 |
| LDL cholesterol (mmol l⁻¹), median [Q1, Q3] | 1.8 [1.4, 2.4] | 1.9 [1.4, 2.4] | 1.8 [1.4, 2.4] |
| Triglycerides (mmol l⁻¹), median [Q1, Q3] | 1.4 [1.0, 1.8] | 1.4 [1.1, 1.9] | 1.3 [0.9, 1.7] |
| HbA1C, median [Q1, Q3] | 38.0 [36.0, 42.0] | 38.0 [35.0, 43.0] | 39.0 [37.0, 42.0] |
| Canadian Cardiovascular Society angina class | | | |
| Class 0: asymptomatic angina, *n* (%) | 0 (0.0%) | 0 (0.0%) | 0 (0.0%) |
| Class I: angina with strenuous exertion, *n* (%) | 28 (11.2%) | 14 (11.1%) | 14 (11.3%) |
| Class II: angina with moderate exertion, *n* (%) | 128 (51.2%) | 63 (50.0%) | 65 (52.4%) |
| Class III: angina with mild exertion, *n* (%) | 89 (35.6%) | 47 (37.3%) | 42 (33.9%) |
| Class IV: angina at rest, *n* (%) | 5 (2.0%) | 2 (1.6%) | 3 (2.4%) |
| Rose Angina Questionnaire | | | |
| Rose Angina, *n* (%) | 135 (54.0%) | 66 (52.4%) | 69 (55.6%) |
| Rose Angina grade I, *n* (%) | 74 (55.2%) | 42 (64.6%) | 32 (46.4%) |
| Rose Angina grade II, *n* (%) | 60 (44.8%) | 23 (35.4%) | 37 (53.6%) |
| SAQ | | | |
| SAQ summary, mean (s.d.) | 51.1 (20.7) | 53.0 (20.9) | 49.2 (20.4) |
| SAQ angina frequency, mean (s.d.) | 55.8 (25.5) | 58.3 (25.3) | 53.4 (25.5) |
| SAQ physical limitation, mean (s.d.) | 53.6 (25.0) | 53.4 (24.3) | 53.9 (25.8) |
| SAQ quality of life, mean (s.d.) | 43.5 (23.7) | 46.4 (25.5) | 40.5 (21.6) |
| SAQ stability, mean (s.d.) | 51.1 (26.0) | 51.2 (26.2) | 51.0 (25.8) |
| SAQ treatment satisfaction, mean (s.d.) | 66.6 (23.1) | 68.7 (22.5) | 64.4 (23.6) |
| Quality of life (EQ-5D-5L) | | | |
| Index score, mean (s.d.) | 0.6 (0.3) | 0.6 (0.2) | 0.6 (0.3) |
| Visual Analogue Scale, median [Q1, Q3] | 60.0 [45.0, 75.0] | 60.0 [49.2, 75.0] | 60.0 [40.0, 75.0] |
| Non-invasive diagnostic stress testing | | | |
| Any stress testing performed, *n* (%) | 173 (69.2%) | 87 (69.0%) | 86 (69.4%) |
| Treadmill stress testing performed, *n* (%) | 151 (87.3%) | 74 (85.1%) | 77 (89.5%) |
| Abnormal treadmill stress testing results, *n* (%) | 80 (53.0%) | 36 (48.6%) | 44 (57.1%) |
| Stress SPECT performed, *n* (%) | 23 (13.3%) | 14 (16.1%) | 9 (10.5%) |
| Abnormal stress SPECT results, *n* (%) | 11 (47.8%) | 6 (42.9%) | 5 (55.6%) |
| Stress ECG performed, *n* (%) | 4 (2.3%) | 2 (2.3%) | 2 (2.3%) |
| Abnormal stress ECG results, *n* (%) | 3 (75.0%) | 2 (100.0%) | 1 (50.0%) |
| CCTA | | | |
| CCTA performed, *n* (%) | 39 (15.6%) | 22 (17.5%) | 17 (13.7%) |
| CCTA normal, *n* (%) | 5 (12.8%) | 4 (18.2%) | 1 (5.9%) |
| Intermediate coronary artery disease, *n* (%) | 20 (51.3%) | 12 (54.5%) | 8 (47.1%) |
| Obstructive coronary artery disease, *n* (%) | 14 (35.9%) | 6 (27.3%) | 8 (47.1%) |

Values are mean (s.d.), median [Q1, Q3] or *n* (%). ª QRISK3 score. Hb, hemoglobin; HDL, high-density lipoprotein; LDL, low-density lipoprotein; SPECT, single-photon emission computed tomography.

12 months. Blinding effectiveness at 12 months after randomization was prospectively assessed and confirmed in all (*n* = 250) participants (Supplementary Table 2).

The SAQ summary score improved at 6 months and 12 months to a greater extent in the intervention group compared to the control group (Table 4 and Supplementary Figs. 1 and 2). At 6 months, the mean ± s.d. SAQ summary scores in the intervention and control groups were 67.3 ± 21.7 (18.1 ± 19.0 change from baseline) versus 53.7 ± 23.4 (0.7 ± 19.8 change from baseline) (adjusted mean difference: 15.6 (95% confidence interval: 11.1–20.2; *P* < 0.001)) (Table 4), and these differences were also evident at 12 months (70.9 ± 23.6 (21.7 ± 22.6 change from baseline) versus 52.1 ± 24.1 (−0.8 ± 20.4 change from baseline))

(adjusted mean difference: 20.9 (95% confidence interval: 15.8–26.0)). This was consistent across all SAQ domains at 6 months (Table 4), including angina limitation (*P* < 0.001), angina stability (*P* < 0.001), angina frequency (*P* < 0.001), treatment satisfaction (*P* < 0.001) and quality of life (*P* < 0.001). At 12 months, similar improvements were observed for all SAQ domains (*P* < 0.001) (Table 4). A forest plot of standardized intervention effect estimates and 95% confidence intervals on the SAQ scores at baseline to 6 months and 12 months are shown in Supplementary Figs. 7 and 8, respectively.

In a post hoc analysis, we found that the effect of the intervention (disclosure of non-invasive endotyping results) was larger for those patients where non-invasive endotyping led to a different diagnosis

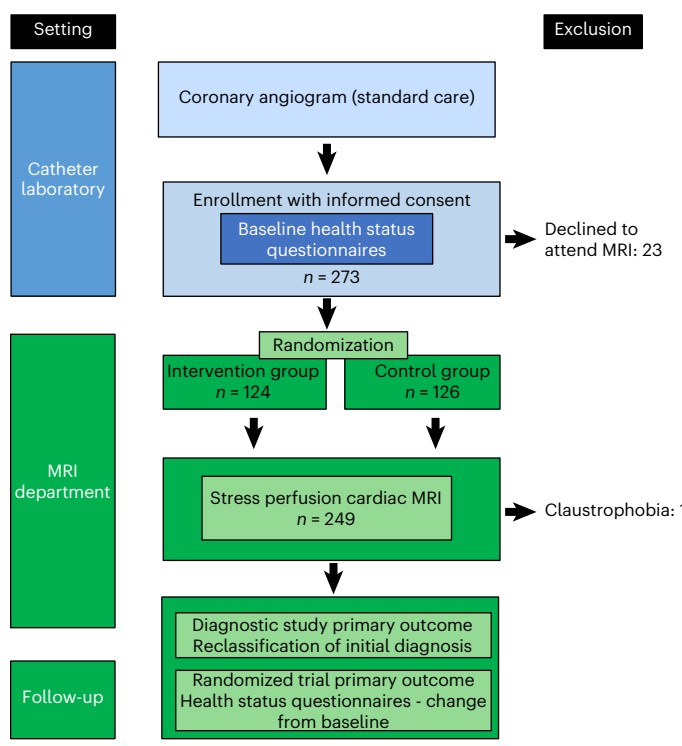

**Fig. 1 | Flow diagram of the clinical trial.** An illustration showing the number of patients enrolled with informed consent in the catheter laboratory, reasons for exclusion, the number of patients in each randomized group and the follow-up process.

compared to angiography (Supplementary Table 3). In those with a different diagnosis, disclosure resulted in an adjusted mean difference in SAQ summary score of 27.48 (95% confidence interval: 21.44−33.52, *P* < 0.001), whereas in those with no change in diagnosis, there was no evidence that disclosure affected SAQ summary score (adjusted mean difference 3.43 (95% confidence interval: −2.88 to 9.75), *P* = 0.285, $P_{interaction}$ < 0.001). The magnitude of this effect was even greater at 12 months (Table 3).

Improvements were also observed in health-related quality of life (as assessed by the EQ-5D-5L instrument) at these timepoints (adjusted mean difference at 6 months: 0.050 (95% confidence interval: 0.00−0.10); at 12 months: 0.09 (95% confidence interval: 0.04−0.13)) (Supplementary Table 4 and Supplementary Figs. 3 and 5). Similar improvements were observed for the EQ-5D-5L visual analogue score (Supplementary Table 4 and Supplementary Figs. 4 and 6). A forest plot of standardized intervention effect estimates and 95% confidence intervals on the EQ-5D scores at baseline to 6 months and 12 months are shown in Supplementary Figs. 7 and 8, respectively.

### Medical management
Medications prescribed in the randomized population at baseline and 12 months are described in Supplementary Table 5. In the intervention group compared to in the control group, in keeping with the higher proportions of patients diagnosed with ANOCA endotypes, participants were more frequently prescribed anti-anginal therapy at the final follow-up visit (83.9% versus 64.3%; *P* = 0.001) and more frequently prescribed calcium channel blockers (43.5% versus 27.0%; *P* = 0.008) and long-acting nitrates (56.5% versus 32.5%; *P* < 0.001). There were similar rates of prescription of beta-blockers (46.0% versus 46.8%).

At the final follow-up visit, more participants in the intervention group were prescribed preventive therapy (85.5% versus 67.5%; *P* = 0.001). Participants in the intervention group were more frequently

prescribed aspirin (77.4% versus 56.3%; *P* < 0.001) and statins (81.5% versus 61.9%; *P* = 0.001).

### Cardiovascular risk factors
Cardiovascular risk factors are described in Table 5. At the final follow-up visit, systolic blood pressure was lower in the intervention group (132.0 mmHg) compared to the control group (140.1 mmHg), with a statistically significant difference in change in systolic blood pressure compared to baseline (−9.57 (95% confidence interval: −13.31 to −5.84) mmHg, *P* < 0.001). More patients in the intervention group had a systolic blood pressure lower than 130 mmHg at follow-up (66 (53.2%) versus 36 (29.0%)).

At the final follow-up visit, body mass index (BMI) was lower in the intervention group (29.5 kg m$^{-2}$) compared to the control group (30.1 kg m$^{-2}$), with a statistically significant difference in change in BMI compared to baseline (−0.80 (95% confidence interval: −1.45 to −0.16), *P* = 0.015). At follow-up, waist circumference was also lower in the intervention group (98.2 cm) compared to the control group (99.6 cm), with a statistically significant change in waist circumference compared to baseline (−2.79 (95% confidence interval: −4.81 to −0.76), *P* = 0.007). There were no differences in blood lipids between the two groups.

At 12 months, predicted 10-year cardiovascular risk, as measured by SCORE2 (ref. 16) and SCORE2 Older Persons (SCORE2-OP)[17], was lower in the intervention group (6.2%) than in the control group (7.6%), with a statistically significant reduction in 10-year cardiovascular risk compared to baseline (−0.89 (95% confidence interval: −1.44 to −0.35), *P* = 0.001).

### Clinical outcomes
**Incidental findings revealed by CMR imaging.** Incidental findings revealed by CMR imaging occurred in four participants (Supplementary Table 6). Two participants had a change in diagnosis from non-cardiac chest pain after angiography to myocarditis, and two participants had a change in diagnosis from non-cardiac chest pain to hypertrophic cardiomyopathy.

**Adverse events and CMR imaging.** No serious adverse events occurred in relation to stress CMR imaging. One participant, a 61-year-old female with a history of type 2 diabetes mellitus and migraine, experienced spontaneous angina during the rest phase of the CMR examination. This was judged to be secondary to psychological stress associated with undergoing the scan and consistent with probable vasospastic angina. Accordingly, treatment with bisoprolol 2.5 mg daily was changed to a long-acting formulation of diltiazem 200 mg once daily.

### Post-discharge clinical outcomes
Vital status and episodes of secondary care were obtained for all participants. Clinical events are described in Supplementary Table 7. No statistically significant difference was observed in clinical outcomes between the two groups. One patient in the control group experienced a non-fatal myocardial infarction. There were no deaths during the follow-up period.

### Clinical cases
Two illustrated clinical cases are provided in Supplementary Figs. 9 and 10.

## Discussion
In this randomized, controlled trial, disclosure of myocardial blood flow by stress CMR imaging undertaken in patients with chest pain and no obstructive coronary artery disease improved diagnosing the cause of angina. Endotyping-informed therapy in the intervention group led to improvements in angina burden and health-related quality of life. After standard coronary angiography, microvascular angina was

**Table 2 | Non-invasive coronary endotyping in the randomized population**

| | All | Control | Intervention | P value |
|---|---|---|---|---|
| | *n*=250 | *n*=126 | *n*=124 | |
| CMR imaging | | | | |
| CMR performed, *n* (%) | 249 (99.6%) | 125 (99.2%) | 124 (100.0%) | |
| Left ventricular ejection fraction (%), mean (s.d.) | 63.4 (5.9) | 63.9 (6.2) | 62.9 (5.6) | |
| Abnormal late gadolinium enhancement, *n* (%) | 42 (16.8%) | 16 (12.7%) | 26 (21.0%) | |
| Ischemic late gadolinium enhancement, *n* (%) | 18 (42.9%) | 5 (31.2%) | 13 (50.0%) | |
| Non-ischemic late gadolinium enhancement, *n* (%) | 24 (57.1%) | 11 (68.8%) | 13 (50.0%) | |
| Myocardial perfusion imaging | | | | |
| Global myocardial blood flow during stress (ml min$^{-1}$ g$^{-1}$), mean (s.d.) | 2.3 (0.6) | 2.2 (0.6) | 2.3 (0.6) | |
| Global myocardial perfusion reserve, mean (s.d.) | 2.9 (0.8) | 2.9 (0.8) | 2.9 (0.7) | |
| Endocardial myocardial perfusion reserve, mean (s.d.) | 2.6 (0.7) | 2.6 (0.7) | 2.7 (0.7) | |
| Coronary microvascular dysfunction | | | | |
| Global myocardial blood flow under stress <2.25, *n* (%) | 131 (52.6%) | 68 (54.4%) | 63 (50.8%) | |
| Global myocardial perfusion reserve <2.2, *n* (%) | 45 (18.1%) | 26 (20.8%) | 19 (15.3%) | |
| Endocardial myocardial perfusion reserve <2.41, *n* (%) | 103 (41.4%) | 51 (40.8%) | 52 (41.9%) | |
| Post-angiogram, pre-randomization endotype | | | | P=0.683 |
| Obstructive coronary artery disease, *n* (%) | 2 (0.8%) | 1 (0.8%) | 1 (0.8%) | |
| Microvascular angina, *n* (%) | 4 (1.6%) | 1 (0.8%) | 3 (2.4%) | |
| Vasospastic angina, *n* (%) | 0 (0.0%) | 0 (0.0%) | 0 (0.0%) | |
| Non-cardiac chest pain, *n* (%) | 244 (97.6%) | 124 (98.4%) | 120 (96.8%) | |
| Other, *n* (%) | 0 (0.0%) | 0 (0.0%) | 0 (0.0%) | |
| Post-angiogram, post-randomization, post-CMR endotype | | | | P=0.905 |
| Obstructive coronary artery disease, *n* (%) | 0 (0.0%) | 0 (0.0%) | 0 (0.0%) | |
| Microvascular angina, *n* (%) | 127 (51.0%) | 65 (52.0%) | 62 (50.0%) | |
| Vasospastic angina, *n* (%) | 1 (0.4%) | 1 (0.8%) | 0 (0.0%) | |
| Non-cardiac chest pain, *n* (%) | 117 (47.0%) | 57 (45.6%) | 60 (48.4%) | |
| Other, *n* (%) | 4 (1.6%) | 2 (1.6%) | 2 (1.6%) | |
| Final diagnosis after randomization | | | | P<0.001 |
| Obstructive coronary artery disease, *n* (%) | 1 (0.4%) | 1 (0.8%) | 0 (0.0%) | |
| Microvascular angina, *n* (%) | 63 (25.2%) | 1 (0.8%) | 62 (50.0%) | |
| Vasospastic angina, *n* (%) | 0 (0.0%) | 0 (0.0%) | 0 (0.0%) | |
| Non-cardiac chest pain, *n* (%) | 184 (73.6%) | 124 (98.4%) | 60 (48.4%) | |
| Other, *n* (%) | 2 (0.8%) | 0 (0.0%) | 2 (1.6%) | |
| Diagnosis reclassified (%), 95% confidence interval | 52.6% (46.2–58.9%) | | | |

*P* values are all two-sided and are from Fisher's exact test for categorical variables or the Mann–Whitney *U*-test for continuous variables.

commonly misdiagnosed as non-cardiac chest pain. The study results apply equally by sex.

Novel design features of this clinical trial included prospective multicenter recruitment; use of validated questionnaires; quantification of myocardial blood flow by stress CMR imaging performed in a single reference center; randomization before CMR imaging; a control procedure; blinding of participants, research staff and clinicians; and stratified medical therapy. Follow-up was complete in all of the participants.

In this population, microvascular angina was common, and, in general, angina severity was moderately high. In the control group, most (51.6%) patients would have had a different diagnosis had the CMR imaging results been disclosed (Tables 2 and 3). In fact, 65 (98.5%) participants in the control group had a final diagnosis of microvascular or vasospastic angina, having been initially diagnosed as having

non-cardiac chest pain based on angiography-guided management. This potential misdiagnosis rate is consistent with previous studies involving invasive testing[10,11] or rubidium-82 myocardial perfusion positron emission tomography–computed tomography[18] (42%) but higher than observations from systematic reviews (41–43%[3,19]). Overall, the results highlight the diagnostic gap for microvascular angina using coronary angiography without adjunctive functional tests.

Stratified medicine is the identification of key subgroups of patients (strata) within a heterogeneous population—these patient strata being distinguishable by distinct mechanisms of disease and/or responses to therapy (endotype)[20]. In this outpatient population, stratified medicine led to improvements in angina and quality of life. This result is consistent with findings in the CorMicA[10] and ILIAS ANOCA[11] clinical trials. In the CorCTCA trial[12], medical management was disrupted during the pandemic, and patients' angina symptoms and

**Table 3 | Differences between original diagnosis (based on clinical history and invasive coronary angiogram) and final diagnosis based on additional information after CMR imaging**

| | | Post-CMR imaging diagnosis | | | | | Final working diagnosis (control group) |
|---|---|---|---|---|---|---|---|
| | | Obstructive CAD | Microvascular angina | Vasospastic angina | Non-cardiac chest pain | Other diagnosis | |
| Control *n*=126 | Original diagnosis | | | | | | |
| | Obstructive CAD | 0 | 1 | 0 | 0 | 0 | **1 (0.8%)** |
| | Microvascular angina | 0 | 1 | 0 | 0 | 0 | **1 (0.8%)** |
| | Vasospastic angina | 0 | 0 | 0 | 0 | 0 | **0 (0.0%)** |
| | Non-cardiac chest pain | 0 | 63 | 1 | 57 | 2 | **124 (98.4%)** |
| | Other | 0 | 0 | 0 | 0 | 0 | **0 (0.0%)** |
| | | 0 (0.0%) | 65 (51.6%) | 1 (0.8%) | 57 (45.2%) | 2 (1.6%) | |
| | | Obstructive CAD | Microvascular angina | Vasospastic angina | Non-cardiac chest pain | Other diagnosis | |
| Intervention *n*=124 | Original diagnosis | | | | | | |
| | Obstructive CAD | 0 | 1 | 0 | 0 | 0 | 1 (0.8%) |
| | Microvascular angina | 0 | 1 | 0 | 2 | 0 | 3 (2.4%) |
| | Vasospastic angina | 0 | 0 | 0 | 0 | 0 | 0 (0.0%) |
| | Non-cardiac chest pain | 0 | 60 | 0 | 58 | 2 | 120 (96.8%) |
| | Other | 0 | 0 | 0 | 0 | 0 | 0 (0.0%) |
| | **Final working diagnosis (intervention group)** | **0 (0.0%)** | **62 (50.0%)** | **0 (0.0%)** | **60 (48.4%)** | **2 (1.6%)** | |

The 'final working diagnosis' (bold) was used to guide treatment decisions after randomization. CAD, coronary artery disease.

health-related quality of life did not improve, highlighting the importance of continuity of care for patients with angina. In the present trial, the improvements in angina and health-related quality of life observed over 1 year in the functional imaging-guided group may reflect an improved understanding of medical therapy for microvascular angina and continuity of care after the pandemic.

The results of our study have implications for clinical practice. Guidelines state that 'Functional imaging for myocardial ischemia is recommended if CCTA has shown CAD of uncertain functional significance or is not diagnostic' (class I, level of evidence B)[1]. The guidelines also state[1] that 'Invasive coronary angiography with the availability of invasive functional assessments is recommended to confirm or exclude the diagnosis of obstructive coronary artery disease or ANOCA/INOCA in individuals with an uncertain diagnosis on non-invasive testing' (class I, level of evidence B) and that 'In persistently symptomatic patients despite medical treatment with suspected ANOCA/INOCA and poor quality of life, invasive coronary functional testing is recommended to identify potentially treatable endotypes and to improve symptoms and quality of life, considering patient choices and preferences' (class I, level of evidence B)[1]. North American guidelines provide similar recommendations[2]. However, during daily practice, coronary function tests to evaluate suspected ANOCA are not routinely used with either invasive angiography (only 7.6% of participants in our study) or CCTA. In the Scottish Computed Tomography of the Heart trial[21], compared to standard care based on functional testing, an anatomical strategy using CCTA added to standard care was associated with relatively worse angina symptoms and health-related quality of life[22]. Based on results from observational studies of non-invasive[3] and invasive functional tests[23–25] and randomized trials[10,11,26], the results from our trial provide new evidence in support of a primary role for functional coronary angiography—that is, functional imaging coupled with either invasive angiography or CCTA. The findings also indicate that patient benefits may be achieved from a functional strategy that does not include intra-coronary acetylcholine testing.

Management guided by stress CMR imaging led to improvements in modifiable risk factors, such as blood pressure and BMI. The stratified medicine approach reduced systolic blood pressure and the proportion of patients with systolic hypertension (systolic >130 mmHg). The effect on systolic blood pressure may be explained by enhanced prescription of angina medication with blood-pressure-lowering effects in the intervention group and a rise in blood pressure (3.7 mmHg) in the control group (Table 5). Although the clinical outcomes at 6 months were not different between the groups, the 10-year predicted risk of myocardial infarction or stroke was also lowered in the intervention group.

Medical management was implemented by blinded clinicians in primary and secondary care. This design minimizes bias that occurs with an open-label design when unblinded staff implement medical care that may be more intensive in the intervention group than in the control group.

Considering clinical implications, first, coronary microvascular dysfunction was common and underdiagnosed in patients assigned a diagnosis of non-cardiac chest pain, as defined by invasive coronary angiography. Second, routine quantification of myocardial blood flow by stress CMR imaging led to improvements in diagnosing the cause of chest pain and changes in guideline-directed medical therapy and improvements in angina burden and health-related quality of life at 6 months and 12 months. No adverse events occurred in relation to the CMR scan. Finally, half of the participants in this trial were female, underlining that females are commonly affected by ANOCA, with implications for quality of life and morbidity.

Considering limitations, coronary function testing was not routinely undertaken during routine care, which appears to reflect contemporary practice more generally. Angina due to coronary spasm can be accurately assessed only by invasive acetylcholine testing, and this approach should be considered for patients with refractory symptoms[1]. Although stress CMR imaging is not widely available, nuclear stress imaging is more accessible, and the diagnostic management of patients with stable chest pain by CMR or nuclear imaging is associated with similar outcomes[27]. Because intra-coronary acetylcholine testing was not included, the post-angiogram, post-randomization, post-CMR endotype diagnosis of 'non-cardiac chest pain' may have included missed diagnoses of vasospastic angina.

In conclusion, half of this outpatient population with chest pain had evidence of coronary microvascular dysfunction, and angina severity was moderately high. Non-invasive endotyping by myocardial blood

**Table 4 | Primary outcome for the randomized trial: SAQ model results**

| Timepoint | Control n=126 | | Intervention n=124 | | Original model: estimate (95% CI), P value | Reduced model: estimate (95% CI), P value |
|---|---|---|---|---|---|---|
| | Follow-up value | Change from baseline | Follow-up value | Change from baseline | | |
| **SAQ summary score** | | | | | | |
| 6 months Nobs (Nmiss) | 126 (0) | 126 (0) | 124 (0) | 124 (0) | | |
| 6 months mean (s.d.) | 53.7 (23.4) | 0.7 (19.8) | 67.3 (21.7) | 18.1 (19.0) | 15.63 (11.07–20.20), P<0.001 | 16.03 (11.50–20.55), P<0.001 |
| 12 months Nobs (Nmiss) | 126 (0) | 126 (0) | 124 (0) | 124 (0) | | |
| 12 months mean (s.d.) | 52.1 (24.1) | −0.8 (20.4) | 70.9 (23.6) | 21.7 (22.6) | 20.90 (15.82–25.98), P<0.001 | 21.12 (16.10–26.15), P<0.001 |
| **SAQ angina frequency score** | | | | | | |
| 6 months Nobs (Nmiss) | 126 (0) | 126 (0) | 124 (0) | 124 (0) | | |
| 6 months mean (s.d.) | 58.6 (26.6) | 0.3 (28.1) | 72.1 (22.7) | 18.7 (24.2) | 15.06 (9.49–20.63), P<0.001 | 15.68 (10.14–21.22), P<0.001 |
| 12 months Nobs (Nmiss) | 126 (0) | 126 (0) | 124 (0) | 124 (0) | | |
| 12 months mean (s.d.) | 55.4 (27.6) | −2.9 (29.4) | 74.6 (23.6) | 21.2 (27.1) | 20.79 (14.78–26.79), P<0.001 | 21.11 (15.17–27.04), P<0.001 |
| **SAQ physical limitation score** | | | | | | |
| 6 months Nobs (Nmiss) | 119 (7) | 114 (12) | 120 (4) | 117 (7) | | |
| 6 months mean (s.d.) | 55.4 (27.7) | 2.0 (19.1) | 68.2 (26.4) | 14.2 (19.9) | 11.79 (6.88–16.70), P<0.001 | 12.21 (7.34–17.09), P<0.001 |
| 12 months Nobs (Nmiss) | 118 (8) | 114 (12) | 120 (4) | 118 (6) | | |
| 12 months mean (s.d.) | 54.0 (26.9) | 0.5 (19.0) | 72.0 (27.8) | 18.3 (24.0) | 17.22 (11.79–22.65), P<0.001 | 17.69 (12.34– 23.04), P<0.001 |
| **SAQ quality of life score** | | | | | | |
| 6 months Nobs (Nmiss) | 126 (0) | 126 (0) | 124 (0) | 124 (0) | | |
| 6 months mean (s.d.) | 47.4 (24.6) | 0.9 (22.9) | 61.6 (25.2) | 21.1 (23.3) | 17.51 (12.19–22.83), P<0.001 | 17.68 (12.44–22.92), P<0.001 |
| 12 months Nobs (Nmiss) | 126 (0) | 126 (0) | 124 (0) | 124 (0) | | |
| 12 months mean (s.d.) | 46.0 (26.4) | −0.4 (24.3) | 66.4 (26.6) | 25.9 (26.4) | 23.60 (17.72–29.47), P<0.001 | 23.63 (17.83–29.44), P<0.001 |
| **SAQ stability score** | | | | | | |
| 6 months Nobs (Nmiss) | 126 (0) | 126 (0) | 124 (0) | 124 (0) | | |
| 6 months mean (s.d.) | 51.8 (24.5) | 0.6 (27.7) | 62.9 (24.6) | 11.9 (30.0) | 10.52 (4.71–16.34), P<0.001 | 11.18 (5.44–16.92), P<0.001 |
| 12 months Nobs (Nmiss) | 126 (0) | 126 (0) | 124 (0) | 124 (0) | | |
| 12 months mean (s.d.) | 47.6 (24.8) | −3.6 (31.3) | 61.1 (24.8) | 10.1 (31.2) | 13.52 (7.50–19.54), P<0.001 | 13.51 (7.51–19.51), P<0.001 |
| **SAQ treatment satisfaction score** | | | | | | |
| 6 months Nobs (Nmiss) | 126 (0) | 126 (0) | 124 (0) | 124 (0) | | |
| 6 months mean (s.d.) | 65.7 (23.6) | −3.0 (22.0) | 78.8 (16.7) | 14.4 (25.2) | 14.53 (9.82–19.24), P<0.001 | 14.66 (9.99–19.33), P<0.001 |
| 12 months Nobs (Nmiss) | 126 (0) | 126 (0) | 124 (0) | 124 (0) | | |
| 12 months mean (s.d.) | 62.5 (24.6) | −6.2 (23.8) | 79.9 (19.7) | 15.5 (29.2) | 18.46 (13.08–23.85), P<0.001 | 18.72 (13.41–24.03), P<0.001 |

In each case, a linear regression intervention effect estimate for the 6-month and 12-month value of the score is presented. In the original model, this estimate is adjusted for the baseline value of the score, age, sex, diabetes, prior myocardial infarction, coronary artery disease, left ventricular systolic function and site (lead versus other). In the reduced covariates model, this estimate is adjusted for the baseline value of the score only. CI, confidence interval; Nobs, number observed; Nmiss, number missing.

**Table 5 | Cardiovascular risk factors by randomized group at baseline and follow-up**

| Treatment group | Control | | Intervention | | Original model | Reduced model |
|---|---|---|---|---|---|---|
| | | | | | Estimate (95% CI), P value | Estimate (95% CI), P value |
| | *n*=126 | | *n*=124 | | | |
| Timepoint | Baseline | 12-month follow-up | Baseline | 12-month follow-up | | |
| Systolic blood pressure (mmHg), mean (s.d.) | 136.8 (16.2) | 140.5 (16.8) | 140.1 (19.0) | 132.0 (16.0) | −9.57 (−13.31 to −5.84), P<0.001 | −9.68 (−13.50 to −5.86), P<0.001 |
| Systolic blood pressure <130 mmHg, *n* (%) | 46 (36.5%) | 34 (27.0%) | 36 (29.0%) | 66 (53.2%) | | |
| BMI (kg m⁻²), mean (s.d.) | 30.2 (5.4) | 30.1 (5.7) | 30.5 (4.7) | 29.5 (4.7) | −0.80 (−1.45 to −0.16), P=0.015 | −0.81 (−1.44 to −0.18), P=0.012 |
| BMI <30 kg m⁻², *n* (%) | 66 (52.4%) | 62 (49.2%) | 56 (45.2%) | 65 (52.4%) | | |
| Waist circumference (cm), mean (s.d.) | 96.2 (14.0) | 99.6 (14.3) | 98.1 (12.7) | 98.2 (11.9) | −2.79 (−4.81 to −0.76), P=0.007 | −2.85 (−4.86 to −0.83), P=0.006 |
| Total cholesterol (mmol l⁻¹), mean (s.d.) | 4.1 (1.1) | 4.3 (1.2) | 4.0 (1.0) | 4.1 (1.0) | −0.08 (−0.30 to 0.14), P=0.475 | −0.08 (−0.30 to 0.14), P=0.456 |
| LDL cholesterol (mmol l⁻¹), mean (s.d.)[a] | 2.0 (0.9) | 2.2 (1.0) | 2.0 (0.9) | 2.0 (0.9) | −0.14 (−0.33 to 0.05), P=0.160 | −0.14 (−0.33 to 0.05), P=0.157 |
| Triglycerides (mmol l⁻¹), mean (s.d.) | 1.6 (0.8) | 1.6 (0.9) | 1.4 (0.7) | 1.5 (0.8) | 0.00 (−0.15 to 0.14), P=0.981 | 0.01 (−0.13 to 0.16), P=0.890 |
| SCORE2/ SCORE2-OP-predicted 10-year cardiovascular risk (%), mean (s.d.)[b] | 6.7 (4.7) | 7.6 (5.9) | 6.2 (3.9) | 6.2 (4.2) | −0.89 (−1.44 to −0.35), P=0.001 | −0.86 (−1.42 to −0.31), P=0.002 |

In each case, a linear regression intervention effect estimate for the 6-month and 12-month value of the score is presented. In the original model, this estimate is adjusted for the baseline value of the outcome, age, sex, diabetes, prior myocardial infarction, coronary artery disease, left ventricular systolic function and site (lead versus other). In the reduced covariates model, this estimate is adjusted for the baseline value of the outcome only. [a] LDL-c was calculated at follow-up using the Friedwald equation: LDL-c=total cholesterol−(HDL-c+VLDL-c) where VLDL-c=(triglycerides/2.2), with all measured in mmol l⁻¹ and converted to mg dl⁻¹=18×mmol l⁻¹. [b] SCORE2 or SCORE2-OP (≥70 years). CI, confidence interval; HDL-c, high-density lipoprotein cholesterol; LDL-c, low-density lipoprotein cholesterol; VLDL-c, very-low-density lipoprotein cholesterol.

flow imaging improved diagnosing the cause of angina and improved wellbeing. Health economic implications should be assessed.

## Online content

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

Conor P. Bradley[1,2], Gemma McKinley[3], Vanessa Orchard[2], Christina Tiller[4], Beth Stanley[3], Daniel Ang[1,2], Andrew J. Morrow[1,2], Robert Sykes[1,2], Pamela Gildea[2], Maria Petty[2], Richard Brogan[2], David Carrick[5], Damien Collison[2], Hany Eteiba[2], Angie Ghattas[2], Richard Good[1,2], Francis Joshi[2], Mitchell Lindsay[2], Peter McCartney[2], James McGowan[6], Ross McGeoch[5], Keith Robertson[2], Paul Rocchiccioli[2], Aadil Shaukat[2], Stuart Watkins[1,2], Peter Kellman[7], Alex McConnachie[3] & Colin Berry[1,2] ✉

[1]British Heart Foundation Glasgow Cardiovascular Research Centre, University of Glasgow, Glasgow, UK. [2]Golden Jubilee University National Hospital, Clydebank, UK. [3]Robertson Centre for Biostatistics, University of Glasgow, Glasgow, UK. [4]Internal Medicine III, Cardiology & Angiology, Innsbruck Medical University, Innsbruck, Austria. [5]Department of Cardiology, University Hospital Hairmyres, East Kilbride, UK. [6]Department of Cardiology, University Hospital Ayr, Ayr, UK. [7]National Heart, Lung, and Blood Institute, National Institutes of Health, Bethesda, MD, USA. ✉e-mail: colin.berry@glasgow.ac.uk

## Methods

### Protocol versions

The original protocol (version 1.0) was approved by the Research Ethics Committee on 11 January 2020.

It subsequently underwent minor amendments, and protocol version 2.0 was approved by the Research Ethics Committee on 21 January 2022. Minor amendments to exploratory outcomes included:

- Remove Fibromyalgia Survey Questionnaire (FSQ)
- Include iMTA Productivity Cost Questionnaire (iPCQ) and symptom log
- Electronic case report form revision to include COVID-19 history
- References update

### Study design

This was a prospective, multicenter, parallel group, 1:1 randomized, controlled superiority trial[28]. The protocol was developed through peer review including the involvement of individuals with lived experience.

The study protocol was approved by the West of Scotland Research Ethics Committee (reference 20/WS/0159) on 11 January 2021, and the protocol was conducted in compliance with the principles outlined in the Declaration of Helsinki. The registration of this study on ClinicalTrials.gov was started on 27 January 2021, was delayed due to record errors and was released by ClinicalTrials.gov on 18 March 2021 (NCT04805814). The reporting of results followed the guidelines outlined in the CONSORT statement. All patients provided written informed consent before study enrollment, demonstrating their understanding of the study procedures and willingness to participate.

Because stress perfusion CMR imaging is not approved for clinical use in National Health Service (NHS) Scotland, the patients who were invited to participate would not otherwise have undergone CMR imaging. At 12 months, the participants were informed of the clinical endotype related to the CMR findings at baseline.

### Population

Potential participants were prospectively identified by having undergone invasive coronary angiography within 3 months, with no obstructive coronary artery disease identified by angiography, and written informed consent[28]. The indication for invasive coronary angiography was 'suspected angina'. The Rose Angina Questionnaire[29] was used to assess symptoms at baseline and define them as typical, atypical or non-anginal.

### Setting

Electronic health records for outpatients referred for assessment of possible coronary artery disease by invasive angiography at three NHS Scotland hospitals (Golden Jubilee University National Hospital, University Hospital Hairmyres and University Hospital Ayr) were screened prospectively.

### Eligibility criteria

The inclusion criteria were as follows:

(1) Age ≥18 years
(2) Symptoms of angina or angina equivalent informed by the Rose Angina Questionnaire
(3) Coronary angiography within 3 months with a plan for medical management

The exclusion criteria were as follows:

(1) Obstructive coronary artery disease—that is, a stenosis >70% in a single segment or 50–70% in two adjacent segments in a coronary artery >2.5 mm or fractional flow reserve ≤0.80
(2) Coronary revascularization by percutaneous coronary intervention or coronary artery bypass graft surgery after the index angiogram
(3) Prior coronary artery bypass surgery
(4) An alternative diagnosis that would explain the angina— for example, anemia, aortic stenosis and hypertrophic cardiomyopathy
(5) Contraindication to contrast-enhanced CMR—for example, estimated glomerular filtration rate <30 ml min$^{-1}$ 1.73m$^{-2}$
(6) Contraindication to intravenous adenosine—that is, severe asthma, long QT syndrome, second-degree or third-degree atrio-ventricular block and sick sinus syndrome
(7) Lack of informed consent

The Study Information Sheet and Consent Form were provided to potentially eligible patients after the standard care coronary angiogram. Patients were invited to participate after the standard care diagnosis had been assigned. Patients who provided written informed consent attended a reference center (Golden Jubilee University National Hospital) for non-invasive endotyping by stress/rest CMR imaging.

### Recruitment

Sex was considered in the study design, and biological sex of the participants was determined by self-reporting. There were no selection or exclusion criteria for adults by age, sex, race or ethnicity.

### Invasive and non-invasive assessment of diagnosis

The initial diagnosis and management plan (usual care) were documented by the interventional cardiologist after coronary angiography and before randomization and CMR imaging. Clinicians and potentially eligible patients were unaware of the possibility of involvement in the research trial at the time of standard care diagnosis. After enrollment, the diagnostic evaluation was coordinated by a research cardiologist (C.P.B.), and the CMR scan was reported by an accredited imaging cardiologist (C.B.).

After coronary angiography, myocardial blood flow was assessed using adenosine stress/rest CMR imaging coupled with in-line pixel mapping in all randomized participants. The imaging cardiologist evaluated the CMR scan for all the participants, and the findings were documented in the research database. In the intervention group, myocardial blood flow findings revealed by CMR imaging were used to inform the final diagnosis described in the radiology report. In the control group, standard angiography-guided care was used to guide management, and the myocardial blood flow findings from CMR imaging were not used to inform the final diagnosis described in the radiology report.

The CMR protocol was identical for all participants. The imaging cardiologist (C.B.), who reported the CMR scan, was blinded to the randomized group. The research cardiologist (C.P.B.), who was not involved in the care of the participants, was unblinded. The intervention involved whether the initial diagnosis (post-coronary angiography) was revised (intervention group—final diagnosis after CMR took account of myocardial blood flow findings) or not (control group— final diagnosis not informed by myocardial blood flow findings). The primary research question was whether coronary angiography and stress perfusion CMR imaging resulted in a different final diagnosis, compared to coronary angiography alone.

Because CMR imaging may disclose prognostically important findings (for example, aortic valve disease, hypertrophic cardiomyopathy and lung mass), clinically relevant incidental findings were disclosed in all patients (independent of the randomized group) and actioned in line with standard care. This approach did not lead to unblinding.

The radiology report was formatted and communicated into the medical record for all participants as would normally be done in standard care. The standard findings that would normally be included in a radiology report included cardiac dimensions and function, myocardial tissue characterization and late gadolinium enhancement (absent,

present and pattern), and any clinically important incidental findings (all patients, both groups) were described.

The radiology report did not include information on the randomized group or the measurements of myocardial blood flow. In the intervention group, the final diagnosis took account of the myocardial blood flow findings, but the actual measurements were not reported. In the control group, the final diagnosis and related treatment were guided by the angiogram but not by the myocardial blood flow findings.

The radiology report also included a description of any prognostically important finding that was revealed by CMR imaging—for example, aortic stenosis. Therefore, prognostically important incidental findings were disclosed to the attending clinicians and the participants, as would normally be done in standard care. The same endotypes, including microvascular angina and vasospastic angina, could be diagnosed in the control group but without taking account of the myocardial blood flow findings.

## Randomization, groups and blinding

Participants (Extended Data Fig. 1, blue image) who had given consent and attended for CMR imaging were eligible for randomization. Randomization was undertaken when the participant attended for stress CMR imaging, and individuals who initially provided consent but then subsequently withdrew before the CMR scan was scheduled were not randomized. To minimize bias, randomization was performed before the CMR scan, and the participants, radiology technologists and attending healthcare staff responsible for clinical care were not informed of the randomized group allocation or myocardial blood flow quantified by CMR imaging and were, therefore, blinded. A standard radiology report for CMR imaging was provided for all participants, and this report did not disclose the randomized group. The clinical outcome assessors were blinded to randomized group allocation. The effectiveness of blinding was prospectively recorded in the electronic case report form.

A web-based randomization tool was used to assign the participants 1:1 to the intervention group (final diagnosis informed by the stress CMR findings) or the control group (CMR findings not used to influence the final diagnosis). In all participants, the randomized group was not disclosed. Randomization was performed using a minimization algorithm with respect to study site, sex, prior myocardial infarction, evidence of coronary artery disease on angiography, diabetes and left ventricular systolic function (<55%, ≥55%). A small random element was included whereby two randomly selected participants in each block of 10 were allocated at random (one to each arm), to reduce predictability.

## Stress/rest perfusion CMR imaging

CMR imaging was performed at 1.5 Tesla (Siemens, MAGNETOM Avanto) using a standardized protocol. All participants were invited to abstain from caffeine-containing beverages or foodstuffs for 24 hours and from vasoactive medications for 48 hours prior to the CMR examination.

The imaging protocol included localizers, cine imaging for cardiovascular dimensions and function, including long-axis left ventricular imaging (for example, four-chamber and three-chamber acquisitions), aortic cine and flow sequences, short-axis planes through the left ventricle (basal, mid-ventricular and apical) for native myocardial longitudinal relaxation time (T1, ms) mapping and transverse relaxation time (T2, ms) mapping (mid-ventricular only) before intravenous administration of gadolinium contrast media and then followed by intravenous adenosine stress and rest myocardial perfusion imaging, short-axis cine for left ventricular function, late gadolinium enhancement imaging and post-contrast myocardial T1 mapping.

Vasodilator stress was achieved by intravenous infusion of adenosine at a dose of 140 µg kg$^{-1}$ min$^{-1}$ for 4 minutes (increased to 210 µg kg$^{-1}$ min$^{-1}$ for a further 2 minutes if no symptoms or less than 10% heart rate increase). Splenic switch-off was retrospectively confirmed during image analysis to assess for adequate stress. At peak stress, a

gadolinium-based contrast medium (Gadovist; Bayer Healthcare) was injected at 4 ml s$^{-1}$ at a dose of 0.05 mmol kg$^{-1}$. Resting first-pass contrast (0.05 mmol kg$^{-1}$, Gadovist) myocardial perfusion imaging was then performed at least 10 minutes later. An intravenous top-up dose of contrast medium (0.1 mmol kg$^{-1}$, Gadovist) was given to optimize late gadolinium enhancement imaging.

## CMR analysis

The CMR imaging data were reviewed and reported by a Level 3 accredited imaging cardiologist at the Golden Jubilee University National Hospital, as per local standards of care. The CMR analyses were undertaken as per standard reporting guidelines[30] using commercially available software (CVI42; Circle Cardiovascular Imaging). Left ventricular volumes and function were analyzed using manual planimetry. Late gadolinium enhancement was reported (17-segment model) with scores of 0 (no hyperenhancement), 1 (1–25% extent), 2 (26–50%), 3 (51–75%) or 4 (>75%). A positive result was taken as ≥2 adjacent segments (or 60° arc-equivalent if the defect crosses segmental boundaries) with ≥50% transmural extent of ischemia, scar or ischemia–scar combination by protocol. Incidental findings identified during this review were referred for interpretation by a radiologist with further action according to standard care.

Automated quantitative myocardial blood flow mapping was performed using the method described by Kellman et al.[31], including the Gadgetron framework. The method involves a dual-sequence approach for myocardial perfusion acquisition and arterial input function acquisition simultaneously, allowing for quantification of myocardial blood flow (ml min$^{-1}$ g$^{-1}$) for each pixel of left ventricular myocardium. The software allows for automated endocardial and epicardial contouring and segmentation using the American Heart Association 16-segment and 32-segment models. Automated endocardial and epicardial sub-segmentations are achieved by offsetting the epicardial border to 50%. The global myocardial blood flow is automatically calculated by the average of all the pixels and is measured at stress and rest. Automated contouring was reviewed and quality checked by the reporting cardiologist. If errors were noted, automated contouring was removed and replaced by manual contours.

Global myocardial perfusion reserve (MPR) is the ratio of stress to rest myocardial blood flow. MPR can also be calculated specifically for the subendocardial layer (MPR$_{ENDO}$) (calculated by stress MBF$_{ENDO}$ / rest MBF$_{ENDO}$). In keeping with previous studies, in the absence of a regional or visual perfusion defect, a threshold of global stress myocardial blood flow <2.25 ml min$^{-1}$ g$^{-1}$, MPR < 2.2 or MPR$_{ENDO}$ < 2.41 was deemed consistent with a diagnosis of coronary microvascular dysfunction[13,14].

## Definitions of endotypes

The diagnostic groups at the end of the standard care coronary angiogram were obstructive coronary artery disease, microvascular angina, vasospastic angina and non-cardiac chest pain. This diagnosis was established by the attending interventional cardiologist based on the medical history and results of non-invasive tests and the coronary angiogram. Following CMR imaging, the initial diagnosis after angiography was re-evaluated by the imaging cardiologist for all participants based on the myocardial blood flow and incidental findings.

The myocardial blood flow results were used by the imaging cardiologist to define the true endotype documented in the research database according to diagnostic criteria[13,14]. In the intervention group, these findings were used to revise the final diagnosis and stratify patients into subgroups (endotypes: microvascular angina, vasospastic angina, non-cardiac chest pain and obstructive coronary artery disease). In the control group, the myocardial blood flow results were not used to revise the final diagnosis.

Microvascular angina was defined according to the Coronary Vasomotion Disorders International Study Group (COVADIS) criteria[32]: symptoms of myocardial ischemia, unobstructed coronary arteries

and evidence of microvascular dysfunction. For the purposes of this clinical trial, microvascular dysfunction was defined by myocardial blood flow and MPR criteria based on CMR imaging.

During standard care invasive management, the interventional cardiologist may have performed adjunctive tests of coronary function. A diagnosis of microvascular spasm required provocation and reproduction of anginal symptoms and ischemic echocardiogram (ECG) shifts but no epicardial spasm during acetylcholine testing[32]. Normal test results denoted no obstructive epicardial coronary artery disease (fractional flow reserve (FFR) > 0.80) and no coronary vascular dysfunction (coronary flow reserve (CFR) > 2.0, index of microvascular resistance (IMR) < 25 and negative acetylcholine testing). A diagnosis of vasospastic angina required that three conditions occur during acetylcholine testing: (1) clinically important epicardial vasoconstriction (≥90%), (2) reproduction of the usual chest pain and (3) ischemic changes on the ECG[33].

### Medical management

The healthcare staff with responsibilities for ongoing care received the radiology report, which described the final diagnosis after CMR imaging.

At the end of the CMR scan, after determination of the final diagnosis, the research cardiologist who coordinated the protocol selected a prespecified medical management plan customized for each endotype (supplementary information). This plan was provided for the endotype regardless of the randomized group. The plan involved medical therapy and non-pharmacological (lifestyle) measures to control cardiovascular risk factors according to guideline targets[1,34]. This information was also provided to the primary and secondary care staff with responsibilities for ongoing care.

The patient's primary clinician was encouraged to titrate medications to address persistent symptoms during the follow-up period. The treatment plan was led by the blinded usual care teams rather than the research team, and medication changes were at the discretion of the usual care clinicians. Standardized letters with customized medical management guidelines were sent to the general practitioner and cardiologist with advice on treatment optimization to relieve anginal symptoms (supplementary information). Standard care for patients in the control group consisted of guideline-directed medical therapy. Referral for cardiac rehabilitation was prioritized for patients with a new diagnosis of ischemic heart disease.

Any clinically important incidental finding (cardiac or extra-cardiac) identified during CMR imaging was disclosed, and the participant's management followed standard care for this finding.

### Outcome assessments

The SAQ is a self-administered, disease-specific measure of angina severity that is valid, reproducible and responsive to changes in health-related quality of life[35]. The SAQ quantifies patients' physical limitations caused by angina, the frequency of and recent changes in their symptoms, their satisfaction with treatment and the degree to which they perceive their disease to affect their quality of life. Each scale is transformed to a score of 0 to 100, where higher scores indicate better function (for example, less physical limitation, less angina and better quality of life). The SAQ summary score (scale 0–100; a higher value reflects less angina burden) averages the domains of angina limitation, frequency and quality of life to provide a measure of angina severity[35].

Health status was serially assessed using validated, self-administered questionnaires for quality of life using the EQ-5D-5L. This is a standardized instrument for measuring generic health status whereby higher scores represent better health-related quality of life (on a scale from −0.59 to 1.00)[36].

The reporting timepoints were baseline, 6 months and 12 months, with the latter scheduled as an in-person visit to the clinical research facility.

### Primary outcome

The primary outcome of the diagnostic study was the difference between the initial diagnosis based on coronary angiography versus the final diagnosis after non-invasive endotyping, evaluated in all participants. The primary outcome of the nested randomized trial was change in the SAQ summary score from baseline to 6 months (change at 12 months was a secondary outcome) compared between the disclosure and non-disclosure groups.

Prespecified secondary outcomes of the randomized trial included changes from baseline in the SAQ domain scores and change in health status as measured by the five-level EQ-5D questionnaire, at 6 months and 12 months. Exploratory outcomes included changes from baseline in cardiovascular risk factors (systolic blood pressure, BMI, waist circumference and lipids (total cholesterol, low-density lipoprotein cholesterol and triglycerides) and SCORE2-predicted risk), at 12 months.

### Adjudicated adverse events

Major adverse cardiovascular events were an exploratory outcome. Follow-up assessments for adverse events were performed by research staff who were blinded to the baseline data and randomized groups. The contacts involved in-person visits, telephone follow-up or review of electronic health records. Clinical events identified as potentially relevant were assessed by a Clinical Event Committee. This committee was also blinded to the baseline data and randomized groups. The committee was independent of the investigators, funder and Sponsor.

**Major adverse cardiac and cerebrovascular events.** Major adverse cardiac and cerebrovascular events (MACCE) were defined as the composite of cardiovascular death, non-fatal myocardial infarction, hospitalization for heart failure, non-fatal stroke or transient ischemic attack and resuscitated cardiac arrest or implantable cardiac device. These are spontaneous adverse cardiovascular events.

**Major adverse cardiac events.** Major adverse cardiac events (MACE) were defined as cardiac death, non-fatal myocardial infarction or hospitalization for heart failure. The cardiac MACE were considered for all myocardial infarctions and also for MACE with spontaneous myocardial infarction only (that is, not type 4 or type 5 myocardial infarction, according to the Fourth Universal Definition of Myocardial Infarction).

**Hospital episodes of care for chest pain.** Hospital visits for chest pain episodes (emergency department and chest pain clinic) that may not have led to hospital admission may not fulfil the criteria for a serious adverse event (SAE) but, nonetheless, are episodes of care that are relevant to patients and healthcare providers. These events include, but are not limited to, episodes of adjudicated angina.

### Definitions of adverse events

Adverse event—any untoward medical occurrence in an individual, including occurrences that are not necessarily caused by, or related to, that product.

Adverse reaction—any untoward and unintended response in an individual to an investigational medicinal product that is related to any dose administered to that individual.

### Serious adverse event

Any SAE that

- results in death
- is life-threatening
- requires hospitalization or prolongation of existing hospitalization
- results in persistent or substantial disability or incapacity
- consists of a congenital anomaly or birth defect
- is otherwise considered medically important by the investigator

Other important adverse events/adverse reactions are those that are not immediately life-threatening or do not result in death or hospitalization but may jeopardize the individual or may require intervention to prevent one of the other outcomes listed in the definition above.

## Assessment of adverse events

All adverse events must be assessed for seriousness. SAEs must be assessed for causality, expectedness and severity and notified to the Sponsor. This is the responsibility of the Chief Investigator or designee. SAEs that are potentially relevant to the secondary health outcomes were assessed by the cardiologists who were not involved in the study and were blinded to randomized group allocation and the CMR imaging findings.

## Data management

Data were prospectively recorded in an electronic case database that was custom developed by data managers based in the Robertson Centre for Biostatistics at the University of Glasgow. The database had controlled access that was customized according to the roles (and blinding status) of the designations of the individual staff members.

## Statistical analyses

The study design involved two parts: first, a diagnostic study of coronary endotypes; second, a nested, randomized, controlled trial of the effects of inclusion of the myocardial blood flow results to inform the final diagnosis.

## Primary outcomes of the observational diagnostic study and the randomized trial

For the diagnostic study, we assessed the reclassification rate of the initial diagnosis based on coronary angiography versus the final diagnosis after non-invasive endotyping. This was reported as a percentage with a 95% confidence interval.

For the randomized trial, the change in SAQ summary score was compared between randomized groups using linear regression, adjusting for baseline SAQ summary score and factors used in the minimization algorithm; age was additionally included as a covariate to reduce residual variation. The intervention effect estimate (adjusted between-group mean difference) was reported with a 95% confidence interval and a *P* value.

In a post hoc analysis, for the primary outcome, we tested the interaction between the randomized intervention effect and whether the diagnosis based on coronary angiography differed from the diagnosis based on non-invasive endotyping. The interaction *P* value is reported, along with the within-subgroup intervention effect estimates.

## Secondary outcomes

Secondary outcomes in the randomized trial were analyzed in the same way as the primary outcome. Residual distributions were examined visually, and standard transformations were applied where necessary to improve model fit. Comparisons of other secondary and exploratory outcome variables were done using Fisher tests (categorical outcomes) or the Mann–Whitney *U*-test (continuous outcomes) where appropriate.

## Sample size calculation

The sample size was determined based on the power to detect a clinically relevant difference in the SAQ summary score. If 6-month outcomes could be obtained from 200 patients, the trial would have 80% power to detect a mean between-group difference in SAQ summary score of 0.40 s.d. units. This is a small difference, but we anticipated that not all patients would have their therapy changed as a result of disclosure. Using the myocardial blood flow data for the control (non-disclosure) group, we carried out focused analyses of the subgroup of patients whose therapy might have been altered based on abnormal results. For example, if therapy were altered in 50% of patients, the study would have 80% power to detect a difference in SAQ score of 0.57 s.d. units for these patients; if therapy were altered in 30% of patients, there would be 80% power to detect a between-group difference of 0.74 s.d. units. Allowing for 20% loss to follow-up, 250 participants were needed to undergo stress perfusion CMR imaging.

Statistical analyses were conducted at the data center (Clinical Trials Unit, Robertson Centre for Biostatistics, University of Glasgow) according to a prespecified statistical analysis plan and the intention-to-treat principle. The analyses were conducted using RStudio and R version 4.5.1 (R Foundation for Statistical Computing).

## Inclusion and ethics

This study included local researchers throughout the research process, including the study design, study implementation, data ownership and authorship of publications. The study was designed to be locally relevant and provide generalizable results.

## Reporting summary

Further information on research design is available in the Nature Portfolio Reporting Summary linked to this article.

## Data availability

Data requests will be considered by the Sponsor and the University of Glasgow, which will take account of the scientific rationale, ethics, logistics and resource implications. Data access requests should be initially submitted by email to the Chief Investigator (C.B., corresponding author), and requesters can be expected to wait 4 weeks before the corresponding author informs them of the suitability of the request and if any additional information is required. The Source Data include the deidentified numerical data used for the statistical analyses and deidentified imaging scans (MRI). Data access will be provided through the secure analytical platform of the Robertson Centre for Biostatistics. This secure platform enables access to deidentified data for analytical purposes, without the possibility of removing the data from the server. Requests for transfer of deidentified data (including source imaging scans) will be considered by the University of Glasgow, and, if approved, a collaboration agreement would be expected, taking account of any cost implications. Cost recovery would be expected on a not-for-profit basis.

## Code availability

The statistical code will be available online in GitHub upon publication of this paper: https://github.com/RobertsonCentre/CorCMR.

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

## Acknowledgements

We thank the patients, staff and the funders who supported this study. This study was funded by the British Heart Foundation (RG/F/23/110104 and PG/19/28/34310). The Chief Scientist Office funded the MRI scans. The funders had no role in the planning, implementation, analysis or write-up of the study.

## Author contributions

C.B. conceived and designed the study and wrote the first draft of the manuscript. C.P.B. implemented the protocol and contributed to the first draft. G.M. and A.M. undertook the statistical analyses. C.B. and C.P.B. had full access to the data in the study and take responsibility for its integrity and the data analysis. Other authors edited the manuscript and/or supported the development and implementation of the study protocol.

## Competing interests

R.S. discloses International (WO) Patent Application No. PCT/EP2025/054222 for 'Virtual Coronary Physiology'. D. Carrick has received consultancy fees and conference travel support from Philips and VP Med Group. D. Collison has received speaker/consultancy fees from Abbott and speaker fees/honoraria from GE Healthcare. F.J. received research funding from Boston Scientific and consultancy fees from Boston Scientific, Abbott Vascular and Shockwave Medical. S.W. has received consulting fees from Abbott Vascular. P.K. is a consultant for Siemens Healthcare. C.B. is employed by the University of Glasgow, which holds consultancy and research agreements for his work with Abbott Vascular, AskBio, AstraZeneca, Boehringer Ingelheim, CorFlow, Edwards Lifesciences, Merck, Servier, Novartis, Xylocor and Zoll Medical. None of the other authors has any potential conflicts of interest.

## Additional information

**Extended data** is available for this paper at https://doi.org/10.1038/s41591-025-04044-4.

**Correspondence and requests for materials** should be addressed to Colin Berry.

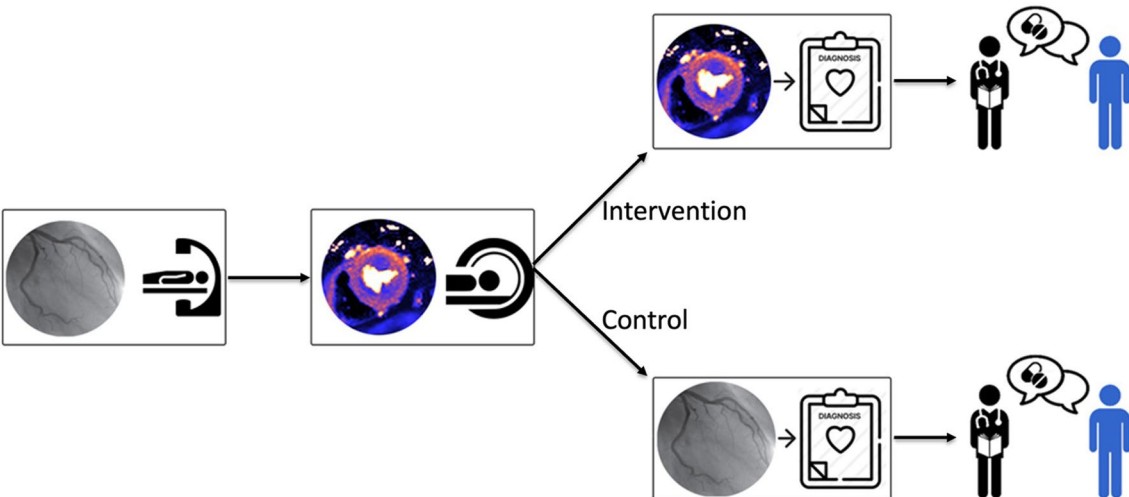

**Extended Data Fig. 1 | Study design and blinding status.** The attending clinicians with responsibilities for ongoing care and the participants were blind to the randomized group allocation and myocardial blood flow results by cardiovascular magnetic resonance (CMR) imaging. The images in this figure include a left coronary angiogram with no evidence of atherosclerosis and a color-encoded map of myocardial blood flow (ml/min/g tissue, range 0 – 6.0). The imaging procedure involves intravenous infusion of adenosine (140 – 210 µg/kg/min) during at least two minutes and when physiological stress is achieved an intravenous bolus of gadolinium-based contrast media (0.05 mmol/kg, Gadovist®) is administered by a pump-injector and the myocardial first-pass of contrast media is then dynamically imaged at 1.5 Tesla field strength (MAGNETOM Avanto Fit, Siemens Healthcare). The quantitation of myocardial blood flow is achieved by an in-line automated pixel-wise myocardial perfusion mapping method[1]. This pixel-map of myocardial blood flow reveals a heterogeneous pattern of hyperemic global myocardial blood flow with a reduction of blood flow in the subendocardium relative to the epicardium. The appearance is consistent with coronary microvascular dysfunction.

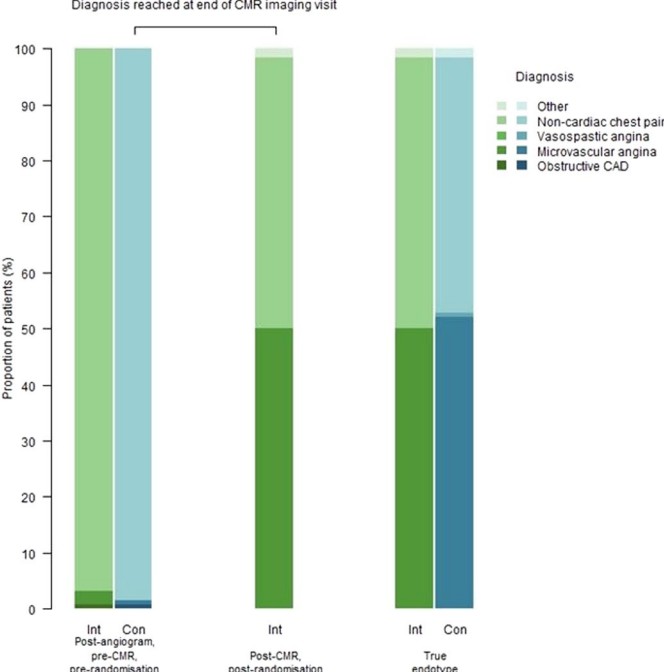

**Extended Data Fig. 2 | Bar chart of diagnoses at sequential timepoints.**
Bar chart of diagnoses at sequential timepoints: post-invasive coronary angiography/pre-CMR imaging (initial diagnosis by invasive angiography), and post-CMR imaging, post-randomization (intervention group, single column, green; myocardial blood flow was acquired in the control group but not used to inform the final diagnosis in the control group). The true endotypes in all patients are also displayed. Colors: intervention group – green, control group – blue. Abbreviations: Con – control, Int – intervention, CAD – coronary artery disease, MRI – magnetic resonance imaging.

# nature research

# Reporting Summary

Nature Research wishes to improve the reproducibility of the work that we publish. This form provides structure for consistency and transparency in reporting. For further information on Nature Research policies, see our Editorial Policies and the Editorial Policy Checklist.

## Statistics

For all statistical analyses, confirm that the following items are present in the figure legend, table legend, main text, or Methods section.

| n/a | Confirmed | |
|---|---|---|
| ☐ | ☒ | The exact sample size (*n*) for each experimental group/condition, given as a discrete number and unit of measurement |
| ☐ | ☒ | A statement on whether measurements were taken from distinct samples or whether the same sample was measured repeatedly |
| ☐ | ☒ | The statistical test(s) used AND whether they are one- or two-sided<br>*Only common tests should be described solely by name; describe more complex techniques in the Methods section.* |
| ☐ | ☒ | A description of all covariates tested |
| ☐ | ☒ | A description of any assumptions or corrections, such as tests of normality and adjustment for multiple comparisons |
| ☐ | ☒ | A full description of the statistical parameters including central tendency (e.g. means) or other basic estimates (e.g. regression coefficient) AND variation (e.g. standard deviation) or associated estimates of uncertainty (e.g. confidence intervals) |
| ☐ | ☒ | For null hypothesis testing, the test statistic (e.g. *F*, *t*, *r*) with confidence intervals, effect sizes, degrees of freedom and *P* value noted<br>*Give P values as exact values whenever suitable.* |
| ☒ | ☐ | For Bayesian analysis, information on the choice of priors and Markov chain Monte Carlo settings |
| ☒ | ☐ | For hierarchical and complex designs, identification of the appropriate level for tests and full reporting of outcomes |
| ☒ | ☐ | Estimates of effect sizes (e.g. Cohen's *d*, Pearson's *r*), indicating how they were calculated |

*Our web collection on statistics for biologists contains articles on many of the points above.*

## Software and code

Policy information about availability of computer code

| Data collection | eEectronic case report form (CRF) developed by programmers in the Robertson Centre for Biostatistics. The eCRF served as a central information repository with restricted access based on centrally administered user rights determined by the chief investigator and coordinated by the Project Management team. The eCRF was developed in line with the protocol. Any changes to the eCRF required sponsor approval. Core laboratory analyses of source data were submitted by site research staff and uploaded directly to the Clinical Trials Unit (CTU) central server. These files were subject to quality assurance procedures administered by data management staff in the CTU. |
|---|---|
| Data analysis | R for Windows v4.5.1 (R Core Team (2021). R: A language and environment for statistical computing. R Foundation for Statistical Computing, Vienna, Austria. URL https://www.R-project.org/.)<br><br>Packages<br>- RODBC: Ripley B, Lapsley M (2023). _RODBC: ODBC Database Access_. R package version 1.3-23,<br>  <https://CRAN.R-project.org/package=RODBC>.<br>- eq5d: Morton F, Nijjar JS (2025). _eq5d: Methods for Analysing 'EQ-5D' Data and Calculating<br>  'EQ-5D' Index Scores_. R package version 0.15.7,<br>  <https://CRAN.R-project.org/package=eq5d>.<br>- readxl: Wickham H, Bryan J (2025). _readxl: Read Excel Files_. R package version 1.4.5,<br>  <https://CRAN.R-project.org/package=readxl>.<br>- dplyr: Wickham H, François R, Henry L, Müller K, Vaughan D (2023). _dplyr: A Grammar of Data<br>  Manipulation_. R package version 1.1.4, <https://CRAN.R-project.org/package=dplyr>.<br>- ggplot2: H. Wickham. ggplot2: Elegant Graphics for Data Analysis. Springer-Verlag New York, 2016.<br>-tidyr survival car QRISK3- tidyr: Wickham H, Vaughan D, Girlich M (2024). _tidyr: Tidy Messy Data_.  doi:10.32614/CRAN.package.tidyr<br><https://doi.org/10.32614/CRAN.package.tidyr>,  R package version 1.3.1,  <https://CRAN.R-project.org/package=tidyr>.<br>- survival: Therneau T (2024). _A Package for Survival Analysis in R_. R package version 3.8-3,  <https://CRAN.R-project.org/package=survival>.<br>Terry M. Therneau, Patricia M. Grambsch (2000). _Modeling Survival Data: Extending the Cox Model_.  Springer, New York. ISBN |

0-387-98784-3.
- car: Fox J, Weisberg S (2019). _An R Companion to Applied Regression_, Third edition. Sage, Thousand Oaks CA. <https://www.john-fox.ca/Companion/>.
- QRISK3: Li Y, Sperrin M, Ltd. C, van Staa TP (2023). _QRISK3: 10-Year Cardiovascular Disease Risk Calculator (QRISK3 2017)_. doi:10.32614/CRAN.package.QRISK3 <https://doi.org/10.32614/CRAN.package.QRISK3>, R package version 0.6.0, <https://CRAN.R-project.org/package=QRISK3>.

For manuscripts utilizing custom algorithms or software that are central to the research but not yet described in published literature, software must be made available to editors and reviewers. We strongly encourage code deposition in a community repository (e.g. GitHub). See the Nature Research guidelines for submitting code & software for further information.

# Data

Policy information about availability of data

All manuscripts must include a data availability statement. This statement should provide the following information, where applicable:

- Accession codes, unique identifiers, or web links for publicly available datasets
- A list of figures that have associated raw data
- A description of any restrictions on data availability

Data were prospectively recorded in an electronic case database that was custom-developed by data managers based in the Robertson Centre for Biostatistics, University of Glasgow. The database had controlled access which was customized according to the roles (and blinding status) of the designations of the individual members of staff.

Code Availability
The statistical code is available online in Github: https://github.com/RobertsonCentre/CorCMR

Anonymised study data will be available on reasonable request by contacting the corresponding author (CB) via the Robertson Centre of Biostatistics. Please allow up to 1 week for a response. Professor Colin Berry, British Heart Foundation Glasgow Cardiovascular Research Centre, School of Cardiovascular and Metabolic Health, 126 University Place, University of Glasgow, Glasgow, G12 8TA, Scotland, UK. Telephone: +44 (0) 141 330 1671 or +44 (0) 141 951 5180. Fax +44 (0) 141 330 6794. Email: colin.berry@glasgow.ac.uk

# Field-specific reporting

Please select the one below that is the best fit for your research. If you are not sure, read the appropriate sections before making your selection.

☒ Life sciences  ☐ Behavioural & social sciences  ☐ Ecological, evolutionary & environmental sciences

For a reference copy of the document with all sections, see nature.com/documents/nr-reporting-summary-flat.pdf

# Life sciences study design

All studies must disclose on these points even when the disclosure is negative.

| Sample size | A pre determined sample size calculation was devised by biostatistician co-authors. The intention-to-treat analysis was the between-group comparison of the reclassification rate using logistic regression, adjusted for baseline factors associated with the likelihood of reclassification of the initial diagnosis with a sample size of 250, the 95% confidence interval of the estimate should have a width of no more than ±6.2%. This should be sufficiently precise to inform the utility of the test. |
| --- | --- |
| | The sample size was determined based on the power to detect a clinically relevant difference in the SAQ summary score. If six month outcomes could be obtained from 200 patients, the trial would have 80% power to detect a mean between-group difference in SAQ summary score of 0.40 standard deviation (SD) units. This is a small difference but we anticipated that not all patients would have their therapy changed as a result of disclosure. Using the myocardial blood flow data for the control (non-disclosure) group, we carried out focused analyses of the sub-group of patients whose therapy might have been altered based on abnormal results. For example, if therapy would be altered in 50% of patients, the study would have 80% power to detect a difference in SAQ score of 0.57 SD units for these patients; if therapy is altered in 30% of patients, there would be 80% power to detect a between-group difference of 0.74 SD units. Allowing for 20% loss-to-follow-up, 250 participants were needed to undergo stress perfusion CMR imaging. |
| Data exclusions | The exclusion criteria were: (1) obstructive coronary artery disease i.e. a stenosis >70% in a single segment or 50 - 70% in two adjacent segments in a coronary artery >2.5 mm, or fractional flow reserve ≤0.80; (2) Coronary revascularization by percutaneous coronary intervention or coronary artery bypass graft surgery following the index angiogram; (3) Prior coronary artery bypass surgery; (4) An alternative diagnosis that would explain the angina e.g. anemia, aortic stenosis, hypertrophic cardiomyopathy; (5) Contra-indication to contrast-enhanced CMR e.g. estimated glomerular filtration rate < 30mL/min/1.73m2; (6) Contra-indication to intravenous adenosine, i.e. severe asthma; long QT syndrome; second- or third-degree atrio-ventricular block and sick sinus syndrome; (7) Lack of informed consent |
| Replication | N/A |
| Randomization | Randomization 1:1 |
| Blinding | Study participants were blind to treatment group and CMR imaging results. |

| Blinding | To minimize bias, randomization was performed before the CMR scan and the participants, radiology technologists and healthcare staff responsible for clinical care were not informed of the randomized group allocation or myocardial blood flow quantified by CMR imaging and were therefore blinded. A standard CMR report was provided for all participants. The report described cardiac mass and function, and clinically significant prognostic findings, e.g. lung mass, but not the results of myocardial blood flow imaging.

The imaging cardiologist (C.Be) who reported the CMR scan was blind to the randomized group. The clinical outcome assessors were blinded to randomized group allocation.

The effectiveness of blinding was prospectively recorded in the electronic case report form. |

# Reporting for specific materials, systems and methods

We require information from authors about some types of materials, experimental systems and methods used in many studies. Here, indicate whether each material, system or method listed is relevant to your study. If you are not sure if a list item applies to your research, read the appropriate section before selecting a response.

## Materials & experimental systems

| n/a | Involved in the study |
|---|---|
| ☒ | ☐ Antibodies |
| ☒ | ☐ Eukaryotic cell lines |
| ☒ | ☐ Palaeontology and archaeology |
| ☒ | ☐ Animals and other organisms |
| ☐ | ☒ Human research participants |
| ☐ | ☒ Clinical data |
| ☒ | ☐ Dual use research of concern |

## Methods

| n/a | Involved in the study |
|---|---|
| ☒ | ☐ ChIP-seq |
| ☒ | ☐ Flow cytometry |
| ☒ | ☐ MRI-based neuroimaging |

## Human research participants

Policy information about studies involving human research participants

| Population characteristics | Potential participants were prospectively identified by having undergone invasive coronary angiography within 3-months, with no obstructive coronary arteries identified by angiography, and written informed consent [16]. The indication for invasive coronary angiography was 'suspected angina'. The Rose angina questionnaire [17] was used to assess symptoms at baseline and define them as typical, atypical, or non-anginal.

Sex was considered in the study design and biological sex of the participants was determined by self-reporting. |
|---|---|
| Recruitment | This study involved a prospective, multicenter, screening and recruitment in individuals.

Electronic health records for outpatients referred for assessment of possible coronary artery disease by invasive angiography at three hospitals in Scotland (National Health Service (NHS) Golden Jubilee National Hospital, University Hospital Hairmyres, and University Hospital Ayr) were screened prospectively.

A screening log was prospectively completed. The reasons for being ineligible, including lack of inclusion criteria and/or presence of exclusion criteria, were recorded. There were no selection or exclusion criteria for adults by age, sex, race or ethnicity.

Eligibility criteria

The inclusion criteria were:
(1) age ≥18 years;
(2) symptoms of angina or angina-equivalent informed by the Rose Angina questionnaire;
(3) coronary angiography ≤3 months with a plan for medical management.

The exclusion criteria were:
(1) obstructive coronary artery disease i.e. a stenosis >70% in a single segment or 50 - 70% in two adjacent segments in a coronary artery >2.5 mm, or fractional flow reserve ≤0.80;
(2) Coronary revascularization by percutaneous coronary intervention or coronary artery bypass graft surgery following the index angiogram;
(3) Prior coronary artery bypass surgery;
(4) An alternative diagnosis that would explain the angina e.g. anemia, aortic stenosis, hypertrophic cardiomyopathy;
(5) Contra-indication to contrast-enhanced CMR e.g. estimated glomerular filtration rate < 30mL/min/1.73m2;
(6) Contra-indication to intravenous adenosine, i.e. severe asthma; long QT syndrome; second- or third-degree atrio-ventricular block and sick sinus syndrome;
(7) Lack of informed consent.

The Study Information Sheet and Consent form were provided to potentially eligible patients after the standard care coronary angiogram. Patients were invited to participate after the standard care diagnosis had been assigned. Patients who |

provided written informed consent attended a reference center (NHS Golden Jubilee hospital) for noninvasive endotyping by stress/rest CMR imaging.

Ethics oversight | The study was approved by the UK National Research Ethics Service (Reference 20/WS/0159).

Note that full information on the approval of the study protocol must also be provided in the manuscript.

# Clinical data

Policy information about clinical studies

All manuscripts should comply with the ICMJE guidelines for publication of clinical research and a completed CONSORT checklist must be included with all submissions.

Clinical trial registration | ClinicalTrials.gov: NCT04900961

Study protocol | The study protocol was peer reviewed and published.

Bradley CP, Orchard V, McKinley G, Heggie R, Wu O, Good R, Watkins S, Lindsay M, Eteiba H, McGowan J, McGeoch R, Corcoran D, Kellman P, McConnachie A, Berry C. The coronary microvascular angina cardiovascular magnetic resonance imaging trial: Rationale and design. Am Heart J. 2023 Nov;265:213-224. doi: 10.1016/j.ahj.2023.08.067. Epub 2023 Aug 30. PMID: 37657593.

Data collection | The study involved three hospitals in central and west Scotland (catchment area, population 2.5 million) - Golden Jubilee National Hospital, University Hospital Hairmyres, and University Hospital Ayr). Two hundred and seventy-three patients were screened and provided informed consent between 9 February 2021 and 18 August 2023 and two hundred and fifty of these patients attended for CMR imaging and were randomized before the scan. Follow-up was continued for 12-months and 100% of the participants complied with the follow-up assessments. There was 0 loss to followup.

Outcomes | The study design involved two parts; first, a diagnostic study of coronary endotypes and second, a nested, randomized, controlled trial of the effects of inclusion of the myocardial blood flow results to inform the final diagnosis.

Primary outcome.

The primary outcome was the between-group difference in the reclassification rate of the initial diagnosis based on coronary angiography versus the final diagnosis after noninvasive endotyping.

Secondary outcomes

Prespecified secondary outcomes included the change in health status as measured by the Seattle Angina Questionnaire (SAQ) summary score and the 5-level EQ-5D questionnaire measured at six and twelve months.

Statistical analyses

Primary outcomes of the observational diagnostic study and the randomized trial

For the diagnostic study, we assessed the reclassification rate of the initial diagnosis based on coronary angiography versus the final diagnosis after noninvasive endotyping. This was reported as a percentage with a 95% confidence interval.

For the randomized trial, the change in SAQ Summary Score was compared between randomized groups using linear regression, adjusting for baseline SAQ SS and factors used in the minimization algorithm; age was additionally included as a covariate to reduce residual variation. The intervention effect estimate (adjusted between-group mean difference) was reported with a 95% confidence interval and p-value.

In a post-hoc analysis, for the primary outcome, we tested the interaction between the randomized intervention effect, and whether the diagnosis based on coronary angiography differed from the diagnosis based on noninvasive endotyping. The interaction p-value is reported, along with the within-subgroup intervention effect estimates.

Secondary outcomes

Secondary outcomes in the randomized trial were analysed in the same way as the primary outcome. Residual distributions were examined visually, and standard transformations were applied where necessary to improve model fit. Comparisons of other secondary and exploratory outcome variables were done using Fisher tests (categorical outcomes) or the Mann-Whitney U test (continuous outcomes) where appropriate.

In preparing this manuscript we have followed the Sex and Gender Equity in Research: rationale for the SAGER guidelines and recommended use. https://researchintegrityjournal.biomedcentral.com/articles/10.1186/s41073-016-0007-6

