## [Peer Review File · Nature Medicine]

Endotyping-Informed Therapy for Patients with Chest Pain and No Obstructive Coronary Artery Disease: A Randomized Trial

Corresponding Author: Professor Colin Berry

Version 0:

Reviewer comments:

Reviewer #1

(Remarks to the Author)

This paper describes a clinical trial that randomized patients with angina of unknown cause to disclosure or not of blood flow findings to the attending cardiologist. Although the patients in the control group underwent the CMR imaging, myocardial blood flow findings were hidden from them for at least 1 year. It would highly surprise me if this study was approved by an Institutional Review Board in the US. It seems obvious that the myocardial blood flow results often change diagnosis (a randomized trial is not needed to make that conclusion), and then doesn't it ethically follow that one must disclose that information? Setting that issue aside, the trial and its statistical analyses seem straightforward and appropriate for the design. I do have some comments that require clarification.

Specific Comments

Page Lines Comment

23 435-437 The list of baseline factors included in the primary intent-to-treat analysis model should be provided. Moreover, the baseline factors are not listed in the SAP. Also, the SAP claims the primary comparison will use "Fisher test", whereas page 23 of the manuscript mentions logistic regression adjusted for baseline factors. The primary secondary outcome (SAQ) is also described as using a baseline-adjusted linear regression model. It is unclear if this means a linear model with only randomized arm and baseline SAQ, or if this model also included other baseline factors. This should be clarified and the list of factors included in the manuscript. The SAP does not list these factors either. In fact, the SAP mysteriously calls SAQ the "primary outcome of the randomized trial" in Section 2.4.1. The SAP is quite confusing as it calls the reclassification rates the primary outcome of the diagnostic cohort study, yet in the same paragraph mentions comparison of the randomized groups.

27 531 It is quite unusual that the comparative result of the primary outcome is not mentioned in this section, but only in the abstract.

44 --- The listed percentage in the Intervention arm with "Diagnosis changed from baseline to microvascular angina" is incorrectly stated at 98.4%. Also, it seems misleading to have "Missed diagnosis" in the table. How can we call anything as missed diagnosis without knowing the truth. If "truth" is considered the CMR result, then the study is even more clearly unethical!

(Remarks on code availability)

Reviewer #2

(Remarks to the Author)

Key results

Patients diagnosed with noncardiac chest pain at angiography often have microvascular disease demonstrated on CMR imaging. Management of microvascular disease improves quality of life and may improve long-term prognosis.

Validity

The trial was well conducted, complete with an elaborate blinding scheme. Management was protocol driven to reduce the introduction of bias. Measurement instruments were objective and appropriate. Sample size and statistical analysis were

appropriate.

Before changing practice, external validity and reproducibility are needed. Additionally, it is unclear how these findings, if reproduced, would be incorporated into management protocols. This will require future work to build on these findings.

Overall I have no critical or major concerns.

Minor concerns and items are listed below.

Significance

These findings are highly significant. Microvascular disease as a cause of chest pain is not novel, nor is diagnosing it via CMR. However, demonstrating that intervening on these patients in a standardized way improves health, in patients who otherwise would have been untreated, has potentially large implications on management of acute chest pain syndromes. This could impact long term survival, healthcare spend, and healthcare utilization.

Data and methodology

The data provided are comprehensive, have adequate methods describing their capture, the appropriate tools were used for quality of life measurement. The specific measurements for myocardial blood flow are outside my scope of expertise. There are some areas that need clarification to ensure this reviewer correctly interpreted the methods, and to ensure readers understand:

1. The distinction between the Radiology report, final diagnosis in the radiology report, and final clinical diagnosis are extremely important and often confusing as written. (lines 227-231 specifically, but also throughout)
2. Lines 200-235 – Please clearly state what was available to form the basis of treatment in each group. All of the information is in these lines, but it required reading numerous times to understand what aspects of information were available in each group. There may actually be too much detail provided.
3. Please provide details linking the treatment protocols to any established guidelines.

Analytical approach

I am not a statistical expert. However, I have no concerns with the analysis approach from a clinician scientist perspective.

Suggested improvements

Major concerns: None

Minor items:

It would be helpful to understand whether the improvements demonstrated in the intervention group were attributable to improvements in patients with microvascular disease or evenly spread across all participants in the intervention group. That would help readers determine whether the pharmacological interventions drove the improvement, or some other unknown variable led to the observed changes.

Please clarify whether most patients were being referred already for CMR after angiography, or were these patients proactively recruited who otherwise would not have undergone CMR.

The sentence from 620-622 is not correctly structured.

Clarity and context

Overall the clarity could be improved. There are areas with excessive detail that obscure the more important big picture of the trial. Such details could be moved to the supplement. Examples: randomization scheme in lines 249-256 could be shortened; lines 306-332 could be grouped conceptually with lines 200-235 and the entire section shortened and clarified. Lines 428-430 are a high level important message about trial design that need to be up-front and emphasized.

References

Appropriate

Your expertise

I am a clinician scientist familiar with CMR clinical trials and the evaluation of patients with acute chest pain, most comfortable with clinical trial design and conduct. I am not an imaging expert nor an expert in statistical analysis.

(Remarks on code availability)

Reviewer #3

(Remarks to the Author)

The authors compared the final diagnosis of patients with angina and no obstructive coronary arteries (ANOCA) on invasive angiography after cardiac MR (CMR). They classified endotypes into microvascular angina, vasospastic angina, obstructive CAD, noncardiac chest pain, and other. Coronary microvascular dysfunction was defined by reduced global myocardial perfusion reserve (MPR), the ratio of stress to rest myocardial blood flow. A threshold of global stress myocardial blood flow $<2.25\text{ml}/\text{min}/\text{g}$, $\text{MPR} < 2.2$, or $\text{MPREND} < 2.41$ was deemed consistent with that diagnosis. They randomized 250 participants with angina and no obstructive disease on invasive angiography who underwent CMR to disclosure of CMR results or non-disclosure. The primary outcome was the reclassification rate of the initial diagnosis (based on coronary angiography) versus the final diagnosis (after noninvasive endotyping). The Seattle Angina Questionnaire (SAQ) summary score and the 5-level EQ-5D questionnaire measured at 6- and 12- months were secondary outcomes. After invasive angiography but before CMR 98% of participants were classified as having non-cardiac chest pain. In the intervention arm, with disclosure of myocardial blood flow findings, 62 (50.0%) had a diagnosis of microvascular angina, a difference of 49.2% (95% CI: 41.3%, 58.7%). A diagnosis of non-cardiac chest pain was predominant in the control arm (98.4%) but much reduced in the intervention arm (48.4%). At six months, SAQ Summary Scores in the intervention and control groups were 67 (18-point improvement from baseline) vs. 54 (0.7-point improvement from baseline). Similar large differences between groups were evident at 12 months.

This is an important study highlighting how frequently we may miss the diagnosis of microvascular angina in patients with

ANOCA. It is well designed and the results are compelling and of great clinical relevance.

1. Please mention that although the trial participants had no obstructive disease, they may have had prior MI and PCI.
2. Why did you not include myocardial bridges as one of the ANOCA endotypes? Were they included in other?
3. Although your study recruited from the cath lab, please speculate in the Discussion as to how this approach to anatomical and functional testing to diagnose coronary microvascular dysfunction might be accomplished with CCTA as the anatomical imaging modality.
4. The differences in medical therapy are substantial, but it is not entirely clear how this was accomplished so effectively. Please provide for the supplement the standardized letters with customized medical management guidelines that were sent to the general practitioner and cardiologist with advice on treatment optimization to relieve anginal symptoms. Please show a template letter for the intervention group and for the control group (if they are different).
5. Medical therapy for microvascular angina is not established, yet you report impressive improvement in the SAQ scores associated with changes in medical therapy. A comment about this might be considered in the Discussion.
6. How many participants from each group participated in cardiac rehabilitation?
7. Please comment on the ability of the average CMR clinical lab to measure myocardial blood flow.

Minor

1. Preventive is preferred use over preventative in text and tables.
2. In Table 1 you wrote "Control" instead of "intervention" in the following: Previous percutaneous coronary Control, n (%)

(Remarks on code availability)

Reviewer #4

(Remarks to the Author)

This study is a randomized control trial among ANOCA patients. The study showed that with noninvasive endotyping, 53.6% of patients were misdiagnosed based on invasive angiography-guided management. In the intervention group, there were improvements in angina burden and health-related quality of life at 6- and 12-months, as assessed by SAQ and EQ-5D-5L. The authors should be commended for the novelty and clinical relevance of this study, which is supported by a well-designed methodology and a thoughtful discussion. References are appropriate. Appropriate use of statistics. Minor comments are provided below.

1. Study design – Good to see that individuals with lived experience were involved in the protocol development.
2. Figure 1 – Further details to this figure are necessary.
3. Line 521-530 – It is stated that no participants had a vasospastic angina diagnosis post-angiogram. However, one participant then had a vasospastic angina diagnosis post-CMR. How did this occur? Did this participant undergo acetylcholine testing and then received a vasospastic angina diagnosis? If so, this needs to be made clearer.
4. Line 616 – is the term 'vasomotor angina' referring to microvascular angina? Or is it being used as a combined term for microvascular angina and vasospastic angina?
5. Line 665 – It is stated that half of the participants in this trial are female which underlines that women are more commonly affected by ANOCA. Is this in reference to coronary artery blockages? This is an unclear statement.
6. Line 393 – It is stated that hospital visits for chest pain episodes were documented as a prioritized outcome of interest. The reader may interpret this as an outcome measure, which is not reported in the results.
7. Limitation section – It is important to note that with the lack of coronary function testing in this study, this may have resulted in mis-diagnoses of vasospastic angina. Therefore, the post-angiogram post-randomization, post-CMR endotype diagnosis of 'non-cardiac chest pain' may include missed diagnoses of vasospastic angina.

(Remarks on code availability)

Version 1:

Reviewer comments:

Reviewer #1

(Remarks to the Author)

Endotyping Patients with Chest Pain and No Obstructive Coronary Artery Disease: A Randomized Controlled Trial

The authors have addressed all of my primary concerns.

I do have some concerns still about Table 3. Last time I questioned the reported percentage for "Diagnosis changed from baseline to microvascular angina" which occurred for 61 out of 124 intervention participants while the reported percentage was 98.4. It seems for some rows in Table 2 you give percentages of a subset of participants, in this case the subset of participants with final diagnosis of microvascular angina. That is confusing and not reflected in the left-hand column describing the row. If you really want to give such percentages of a subset (not recommended), then you need to modify the row description accordingly (for example "Diagnosis of those with final diagnosis of microvascular angina was changed from

baseline”).

Moreover, the numbers in Table 3 don't seem to make any sense to me. Perhaps I don't understand what is meant by reclassified. I see several rows that give numbers of “Diagnosis changed from baseline to X”, which seem to represent disjoint subsets. But when I count up those in the Control and Intervention arms, I get 0 and 65 respectively. Surely this can't be right. Moreover, I then see “Diagnosis reclassified” of 67 and 0 respectively! Nowhere in this table do I see a total (control+intervention) of (the reported) 131 for the primary cohort outcome of reclassification.

(Remarks on code availability)

Reviewer #2

(Remarks to the Author)

The authors have completed a detailed and thorough response to the reviewers' concerns. Their responses are adequate and acceptable to this reviewer. The final manuscript is of high quality and significantly advances the medical literature. This reviewer suggests the manuscript be accepted for publication.

(Remarks on code availability)

Reviewer #3

(Remarks to the Author)

Thank you for your responses to Editor and Reviewer comments. I have no further questions or suggestions.

(Remarks on code availability)

Reviewer #4

(Remarks to the Author)

Thank you for the opportunity to review this manuscript. I am satisfied with the revisions and comments undertaken in response to the reviewer's comments.

(Remarks on code availability)

Code above not working

Response to Referees

Title: Endotyping Patients with Chest Pain and No Obstructive Coronary Artery Disease: A Randomized Controlled Trial

Reference: NMED-A144896

Author response

Dear Dr Messin,

On behalf of the co-authors, I am extremely grateful to the Editors and the Reviewers for kindly considering our research article under Fast Track conditions, and for providing their feedback in such short timelines.

Having considered the Reviewers' comments, we are encouraged that all four reviewers have provided supportive comments on our research, reflecting a consensus. Nonetheless, we also recognise they have also provided important points that require revision of the manuscript.

We have now provided a tracked-change version of the revised manuscript with revised text highlighted in yellow. We have also copy-pasted the revised text in association with our responses below.

Noting your letter, we are greatly encouraged to learn that Nature Medicine is interested in publishing our research. We have very carefully considered the feedback and we have made every effort to revise the manuscript and provide detailed responses (below). We have revised the manuscript in line with the feedback. We have formatted the manuscript in line with Author Guidelines, including the abstract style and word count (150), main text (<6000), reference style, and figures (300 dpi).

Editors Comments

1. Given one of the reviewers (#1) has raised potential ethical concerns we would ask that the authors provide evidence of IRB approval upon resubmission (e.g., a stamped/signed protocol).

RESPONSE: Thank you. We have uploaded the original IRB approval letter. The ethics application was submitted on 19 October 2020 and the committee meeting was scheduled for 6 November 2020 and Colin Berry attended this meeting in person. The committee designation is “NHS West of Scotland Research Ethics Service Committee 4” which is part of the UK National Research Ethics Service.

The application was submitted through the Integrated Research Application System (IRAS): <https://www.myresearchproject.org.uk/Signin.aspx> and the **IRAS Registration No. is 281128**.

On the Nature Medicine website, we have now uploaded the IRB approval letter to “Additional Supplementary Review material”: Approval letter, Research Ethics Committee, dated 11 January 2021 - File name: 20-WS-0159 281128 Ack Add Con 11Jan21.pdf

2. We would also like to know if participants were ever informed of their myocardial blood flow results (i.e., after the 12 month follow-up time).

RESPONSE: Thank you for this question.

Yes, we confirm that all of the participants were informed of their final diagnosis after completing the 12-month visit. In other words, after completion of the responses to the 12-month questionnaires, then at the end of the visit, the final diagnosis was disclosed. In other words, the participants were unaware of the diagnosis until all of the visit data had been recorded.

The implications of the diagnosis were most relevant to participants in the control group and for these patients, their medication was revised in line with the revised diagnosis. If the patient had persisting angina, then follow-up was arranged with the GP and attending cardiologist.

The ‘final diagnosis’ post-randomisation/post-MRI was linked to the myocardial blood flow results. Since the myocardial blood flow (ml/min/g) data were of a technical nature, emphasis was placed on the diagnosis rather than the absolute level of myocardial blood flow, but this was also provided to individual participants if of interest.

3. Also, in line with the concerns of the reviewers we would ask that the authors remove all mention of one diagnosis being "correct" or "incorrect", only that the diagnoses are different. At the time of differential diagnosis it was not clear that the CMR diagnosis is "correct". That is only suggested by the Angina score information.

RESPONSE: This point is well taken. The terms ‘correct’ and ‘incorrect’ have been removed and replaced with ‘Reclassified’ (Table 3; changes marked up in yellow)

4. Please ensure that statistical analyses are performed exactly as they are described in the SAP (see reviewer 1 for more details). The stats between SAP and the manuscript must be in perfect agreement or will we be unable to proceed with the manuscript (unless sufficient justification is provided for the discrepancy).

RESPONSE: Thank you for this feedback, and we apologise for any lack of clarity in the Statistical Analysis Plan (SAP).

For the diagnostic study, the primary outcome is the difference between the original diagnosis (post-angiography, pre-randomisation, pre-CMR) and the final diagnosis (post-CMR) for all participants. The SAP attempted to make this clearer than had been articulated in the protocol.

The primary analysis for the diagnostic study is intended to reveal the magnitude of the impact of CMR imaging on the post-angiography standard care diagnosis, with approximately half of the participants in the Intervention Group receiving a different diagnosis when CMR was used.

We did, however, compare the randomised groups, using Fisher Tests, in three ways (Table 3):

(a) comparing the original diagnosis between randomised groups,

(b) comparing the post-CMR diagnosis between randomised groups, and

(c) comparing the “working” diagnosis (i.e. the original diagnosis for those in the non-disclosure arm, with the post-CMR diagnosis in the disclosure arm) between randomised groups. Comparisons (a) and (b) demonstrate that the randomised groups are similar, in terms of their original and post-CMR diagnoses, as expected given randomisation, whereas comparison (c) shows that the disclosure of the post-CMR diagnosis results in a large difference between groups in the working diagnosis, upon which subsequent treatment decisions were made.

We have updated the manuscript accordingly and detailed information is provided in our response to Reviewer #1.

5. The manuscript divides outcomes as primary (differences in diagnosis) and secondary (angina questionnaire and QoL questionnaire). However, in the SAP these are both defined as primary outcomes (of the observational and randomised parts of the trial). They should be listed as primary outcomes in the manuscript as well.

RESPONSE: Thank you for this feedback which we fully agree with. The SAP reflects how we conceptualized the analysis, treating the diagnostic study and the randomised trial as separate entities, each with its own primary outcome and analysis and the layout of the manuscript reflects the SAP

Statistical analyses / Page 18, line 347 – 364

“Primary outcomes of the observational diagnostic study and the randomized trial

For the diagnostic study, ...

For the randomized trial, the change in SAQ Summary Score was compared ...”

We would like to explain why we arrived with this apparent discrepancy. For background information, for a previous study with a similar design (Sidik N, et al Circulation. 2024 PMID: 37795617), the reviewers preferred a different approach (ie primary outcome = diagnostic study only) and health status (SAQ) as a secondary outcome. Whilst we did not agree with this approach, we had no other option but to do so.

In line with the feedback from the Editor and Reviewer #1, we prefer to follow the approach taken in the SAP and consider these assessments as two, separate, primary outcomes, one for the observational diagnostic study and then angina burden, reflected by the Seattle Angina Questionnaire Summary Score for the randomized, controlled trial. This approach in the manuscript is exactly as described in the SAP.

6. Finally, given that (at present) the manuscript appears retrospectively registered we would ask the authors provide an explanation in the methods section why this is the case. Something similar to what is contained within this manuscript - <https://www.nature.com/articles/s41591-023-02721-w#Sec9>

"The study protocol was approved by the Institutional Review Board of Sun Yat-sen University Cancer Center on 6 June 2019 and the other participating study institution The Sixth Affiliated Hospital of Sun Yat-sen University and was conducted in compliance with the principles outlined in the Declaration of Helsinki. The registration of this study on ClinicalTrials.gov had been started on 21 September 2019, delayed due to record errors, and was released by ClinicalTrials.gov on 10 November 2019 (NCT04250948). The reporting of results followed the guidelines outlined in the CONSORT statement. All patients provided written informed consent before study enrolment, demonstrating their understanding of the study procedures and willingness to participate"

We have no concerns about the registration but we still need it stated clearly in the manuscript.

RESPONSE: Thank you for this feedback and guidance. We appreciate the Editor’s oversight in this matter. We have now edited the manuscript accordingly.

We wish to affirm that the registration application for this study was submitted before the first participant was enrolled. It is clear from the information (upload, with file dates) that we have provided that the study was submitted to Clinicaltrials.gov weeks in advance of the first

participant being enrolled. The Clinicaltrials.gov staff had requested edits to the text of our application. The edits were stylistic and pertained to the standardized language style and terminology required for Clinicaltrials.gov registration.

We are grateful for the Editor's guidance and we have provided the following text in line with the text provided above.

Page 8, lines 130 – 137

“The study protocol was approved by the West of Scotland Research Ethics Committee (reference 20/WS/0159) on 11 January 2021, and the protocol was conducted in compliance with the principles outlined in the Declaration of Helsinki. The registration of this study on ClinicalTrials.gov had been started on 27 January 2021, delayed due to record errors, and was released by ClinicalTrials.gov on 18 March 2021 (NCT04805814). The reporting of results followed the guidelines outlined in the CONSORT statement. All patients provided written informed consent before study enrolment, demonstrating their understanding of the study procedures and willingness to participate.”

We therefore invite you to revise your manuscript taking into account all reviewer and editor comments. Please highlight all changes in the manuscript text file.

*** Include a “Response to referees” document detailing, point-by-point, how you addressed each referee comment. If no action was taken to address a point, you must provide a compelling argument. This response will be sent back to the referees along with the revised manuscript.**

RESPONSE: Thank you. We have responded to all of the reviewers' points in this 'Response to Referees' document.

* If you have not done so already please begin to revise your manuscript so that it conforms to our policy and format instructions here: <https://www.nature.com/nm/submission-guidelines/aip-and-formatting>

RESPONSE: Thank you. We have followed the Nature Submission guidelines.

RESPONSE: Thank you. This is now done.

Nature journals have recently announced an update to our guidance on reporting on sex and gender in research studies (see here). We strongly encourage researchers to follow the ‘Sex and Gender Equity in Research – SAGER – guidelines’ and to include sex and gender considerations for studies involving humans, vertebrate animals and cell lines where relevant to the topic of study (an overview can be found here). Authors should use the terms sex (biological attribute) and gender (shaped by social and cultural circumstances) carefully in order to avoid confusing both terms.

RESPONSE: Thank you. We have used the term ‘sex’ in the manuscript.

Of the overall study population, 50.4% were female, indicating the study is well balanced by distribution of sex and is therefore similarly representative for males and females.

Sex was prespecified for inclusion in the minimization algorithm for randomization (page 15, line 258) and this approach has proven to be effective with reasonably well balanced distribution of females between the control and intervention groups.

Table 1. Baseline Demographic and Clinical Characteristics for the Randomized Population.

Characteristic	All	Control	Intervention
	250	126	124
Female, n (%)	126 (50.4%)	66 (52.4%)	60 (48.4%)

We have made clear in the Reporting Summary (Section Clinical Data / Outcomes):-

In preparing this manuscript we have followed the Sex and Gender Equity in Research: rationale for the SAGER guidelines and recommended use.

<https://researchintegrityjournal.biomedcentral.com/articles/10.1186/s41073-016-0007-6>

When preparing your revised manuscript, please be aware of our guidance on Sex and Gender reporting). Please note that:

- 1. If the research findings apply to only one sex or gender, that must be indicated in the title and/or abstract.**

RESPONSE: Thank you. The research applies equally by sex. We have added this sentence to the Discussion

Page 27, lines 535 – 536

“The study results apply equally by sex.”

2a. For studies involving vertebrates animal and cell lines- The Reporting Summary should include whether sex was were considered in the study design.

RESPONSE: There are no experimental (in vivo or ex vitro) data in our study.

2b. For studies involving human research participants- The Reporting Summary should include whether sex and/or gender was considered in the study design and whether sex and/or gender of participants was determined based on self-report or assigned (and methodology used).

RESPONSE: Thank you for this feedback. In the Reporting Summary, we have described the prespecified considerations for sex.

Page 11, lines 178

Population characteristics

“Sex was considered in the study design and biological sex of the participants was determined by self-reporting.

Recruitment

There were no selection or exclusion criteria for adults by age, sex, race or ethnicity.

3. Data should be reported disaggregated for sex and gender where this information has been collected and consent has been obtained for reporting and sharing individual-level data; disaggregated numbers for individual experiments must be provided in the source data as appropriate whereas overall numbers may be provided in the Nature Portfolio Reporting Summary.

RESPONSE: Thank you for this feedback. We do not have consent for reporting individual disaggregated data since this could lead to inadvertent disclosure of the identify of a participant. The distribution of participants by sex has been described in Table 1.

Information on the 3 points above should be included in the revised manuscript and detailed in the cover letter.

RESPONSE: Thank you. This information has now been provided in pages 1 – 2 of the cover letter.

In addition, please note that if sex- and gender-based analyses have been performed a priori, results should be reported regardless of positive or negative outcome. We discourage conducting post hoc sex- and gender-based analysis if the study design is insufficient (for example, low sample size) to enable meaningful conclusions.

RESPONSE: Thank you for this feedback. We have not undertaken sex-based analyses.

If no sex- and gender-based analyses have been performed, please indicate the reasons for the lack of these analyses in the Reporting Summary.

RESPONSE: Thank you for this feedback.

Yes, sex was a prespecified term that has been taken into account in the evaluation of the primary and secondary outcome analyses for health status, as outlined in the SAP (page 8 and 10).

The following information has been included as a footnote for Table 4 (Seattle Angina Questionnaire, primary outcome of the randomised trial) and Supplementary Table S4 (EuroQol-5D 5 Level health related quality of life)

Page 45

“In each case, a linear regression intervention effect estimate for the 6-months and 12-months value of the score are presented. This estimate is adjusted for the baseline value of the score, age, sex, diabetes, prior myocardial infarction, coronary artery disease, LV systolic function, and site (lead vs other). SD = standard deviation.”

When submitting the revised version of your manuscript, please also pay close attention to our [Digital Image Integrity Guidelines](https://www.nature.com/nature-research/editorial-policies/image-integrity). and to the following points below:

-- that unprocessed scans are clearly labelled and match the gels and western blots presented in figures.

RESPONSE: Thank you for this feedback. Our manuscript does not include data on gels or blots.

-- that control panels for gels and western blots are appropriately described as loading on sample processing controls

RESPONSE: Thank you for this feedback. There are no gels or western blots.

-- all images in the paper are checked for duplication of panels and for splicing of gel lanes.

RESPONSE: Thank you. We confirm there is no duplication of images.

RESPONSE: Thank you. Data are available and accessible at the Robertson Centre for Biostatistics, University of Glasgow. All data are archived in perpetuity.

Page 20, lines 391 – 393

Code Availability Statement

The statistical code will be available online in GitHub on publication of the manuscript:

<https://github.com/RobertsonCentre/CorCMR>

<https://mts-nmed.nature.com/cgi-bin/main.plex?el=A1H2BXa6A2DepS7J6A9ftd4Xot24THpAeZzxjuceIQZ>

We hope to receive your revised manuscript within three to four months. If you cannot send it within this time, please let us know. We will be happy to consider your revision so long as nothing similar has been accepted for publication at Nature Medicine or published elsewhere. Please note that we generally only allow one cycle of major experimental revision and we advise that you take the required time to properly respond to the editorial and reviewers comments.

Nature Medicine is committed to improving transparency in authorship. As part of our efforts in this direction, we are now requesting that all authors identified as ‘corresponding author’ on published papers create and link their Open Researcher and Contributor Identifier (ORCID) with their account on the Manuscript Tracking System (MTS), prior to acceptance. This applies to primary research papers only. ORCID helps the scientific community achieve unambiguous attribution of all scholarly contributions. You can create and link your ORCID from the home page of the MTS by clicking on ‘Modify my Springer Nature account’. For more information please visit www.springernature.com/orcid.

Please do not hesitate to contact me if you have any questions or would like to discuss these revisions further. We look forward to seeing the revised manuscript and thank you for the opportunity to review your work.

Sincerely,

Liam Messin PhD MRSB

Deputy Editor, Nature Medicine

RESPONSE: Thank you. We are very grateful for this timely and comprehensive feedback. We have made every effort to respond proactively in line with the feedback we have received.

Reviewers' Comments:

Reviewer #1 (Remarks to the Author):

This paper describes a clinical trial that randomized patients with angina of unknown cause to disclosure or not of blood flow findings to the attending cardiologist. Although the patients in the control group underwent the CMR imaging, myocardial blood flow findings were hidden from them for at least 1 year. It would highly surprise me if this study was approved by an Institutional Review Board in the US. It seems obvious that the myocardial blood flow results often change diagnosis (a randomized trial is not needed to make that conclusion), and then doesn't it ethically follow that one must disclose that information? Setting that issue aside, the trial and its statistical analyses seem straightforward and appropriate for the design. I do have some comments that require clarification.

RESPONSE: We are very grateful to the Reviewer for highlighting this point which is very well taken.

As highlighted by Reviewer #4, individuals with lived experience were involved in the design of the study protocol. The design was peer reviewed and approved by the Project Grant Committee of the British Heart Foundation:

<https://www.bhf.org.uk/for-professionals/information-for-researchers/how-we-award/project-grants-committee>

The protocol was subsequently reviewed and approved by the West of Scotland Research Ethics Committee 4 on behalf of the UK National Research Ethics Service.
<https://www.hra.nhs.uk/about-us/committees-and-services/res-and-recs/>

The approval letter (reference 20/WS/0159; date of letter 11 January 2021) has been uploaded as 'Additional Review Material' into the Nature Medicine web-portal (file name: 20-WS-0159 281128 Ack Add Con 11Jan21.pdf)

The CorCMR trial is one of a series of clinical trials of diagnostic testing that our group has led during recent years. We wish to highlight:

CorMicA

Ford TJ, Stanley B, Good R, Rocchiccioli P, McEntegart M, Watkins S, Eteiba H, Shaukat A, Lindsay M, Robertson K, Hood S, McGeoch R, McDade R, Yii E, Sidik N, McCartney P, Corcoran D, Collison D, Rush C, McConnachie A, Touyz RM, Oldroyd KG, Berry C. Stratified Medical Therapy Using Invasive Coronary Function Testing in Angina: The CorMicA Trial. J Am Coll Cardiol. 2018 Dec 11;72(23 Pt A):2841-2855. doi: 10.1016/j.jacc.2018.09.006. Epub 2018 Sep 25. PMID: 30266608.

CorCTA

Taqueti VR. Prevalence of Abnormal Coronary Function in Patients With Angina and No Obstructive Coronary Artery Disease on Coronary Computed Tomography Angiography: Insights From the CorCTA Trial. *Circulation*. 2024 Jan 2;149(1):24-27. doi: 10.1161/CIRCULATIONAHA.123.066571. Epub 2023 Dec 28. PMID: 38153994.

In brief, both of the CorMicA and CorCTA trials have been cited in the practice guidelines of the European Society of Cardiology (*Eur Heart J* 2024; PMID: 39210710). These results of these trials have triggered a Class I B recommendation in favour of coronary function testing in patients with “no obstructive coronary arteries (ANOCA/INOCA) and uncertain diagnosis” (Table 13, Class I Level of Evidence B) and “in persistently symptomatic patients despite medical treatment with suspected ANOCA/INOCA” (Table 25, Class I Level of Evidence B) refractory angina and reduced quality of life (Table 13). Initially, the CorMicA trial, published in 2018, led to a Class IIA recommendation by the ESC guidelines for chronic coronary syndromes (Section 6.1.3, *Eur Heart J*, 2020; PMID: 31504439). Prior to the CorMicA trial, no such recommendations existed.

CorMicA was the first randomised, controlled trial of diagnostic strategies in ANOCA. The publication has attracted 631 citations (September 2025):

<https://www.sciencedirect.com/science/article/pii/S0735109718383815?via%3Dihub>

The Reviewers will know that coronary angiography (by noninvasive CTA or invasive) is commonly not associated with functional tests, and in the UK, only CTA is recommended for the evaluation of suspected angina. The results from our new trial highlight that an imaging strategy coupled with coronary angiography leads to clinical benefits, but our trial is the first to provide confirmatory evidence.

Myocardial blood flow imaging by CMR or PET is not available in NHS Scotland, and none of the participants would therefore have had the option of this type of imaging. One reason for CMR and PET not being available is there is no evidence from randomised, trials in this population (suspected noncardiac chest pain post-angiography) that management linked to myocardial blood flow imaging makes any difference to patient wellbeing, and in the absence of evidence, the absolute cost of undertaking these tests rules out access.

Had we undertaken a diagnostic study without randomisation then the study would have been incremental and in line with many prior cross-sectional, observational studies. The key (disruptive) attributes of the study design include randomisation, a sham-control procedure, blinding, longitudinal follow-up, and analysis by blinded statisticians based in a Trials Unit that is independent of the clinical research team. Indeed, these points are highlighted by Reviewer #2.

These design features support conclusions that functional imaging in this population benefits patients, and therefore becomes relevant to healthcare providers on why this strategy would not be undertaken. We also plan to undertake a prespecified health economic analysis.

We are very encouraged that the Reviewer appears to endorse the trial design and statistical analysis. We would also like to highlight that most of the participants in this study had a standard care diagnosis following clinically-indicated angiography of ‘non-cardiac chest pain’.

Specific Comments

Page Lines Comment

23 435-437 The list of baseline factors included in the primary intent-to-treat analysis model should be provided. Moreover, the baseline factors are not listed in the SAP. Also, the SAP claims the primary comparison will use “Fisher test”, whereas page 23 of the manuscript mentions logistic regression adjusted for baseline factors. The primary secondary outcome (SAQ) is also described as using a baseline-adjusted linear regression model. It is unclear if this means a linear model with only randomized arm and baseline SAQ, or if this model also included other baseline factors. This should be clarified and the list of factors included in the manuscript. The SAP does not list these factors either. In fact, the SAP mysteriously calls SAQ the “primary outcome of the randomized trial” in Section 2.4.1. The SAP is quite confusing as it calls the reclassification rates the primary outcome of the diagnostic cohort study, yet in the same paragraph mentions comparison of the randomized groups.

RESPONSE: Thank you for this comment. The baseline characteristics for the intention-to-treat analyses are listed in Table 1.

In relation to the statistics, we thank the Reviewer for highlighting this point and the inconsistency was an oversight in the SAP. All models were adjusted for factors used in the minimisation algorithm, plus the baseline value of the outcome being studied, as per guidelines for RCT analysis. This has been clarified in the methods section of the manuscript. In addition, we adjusted for age in order to reduce residual variation and improve the precision of effect estimates. We have also provided these outputs as ‘reduced covariate models’ (or similar) with the ‘full model’ referred to as the original model. This additional column in each table has been marked up in yellow. Note that all analyses were programmed and validated prior to unblinding of the statistical team and were not influenced by exposure to the randomised groups.

We have clarified the Statistical Methods as being in line with the SAP and we have the revised the text as follows:

Page 18, lines 346 – 364

Primary outcomes of the observational diagnostic study and the randomized trial

For the diagnostic study, we assessed the reclassification rate of the initial diagnosis based on coronary angiography versus the final diagnosis after noninvasive endotyping. This was reported as a percentage with a 95% confidence interval.

For the randomized trial, the change in SAQ Summary Score was compared between randomized groups

using linear regression, adjusting for baseline SAQ SS and factors used in the minimization algorithm; age was additionally included as a covariate to reduce residual variation. The intervention effect estimate (adjusted between-group mean difference) was reported with a 95% confidence interval and p-value.

In a post-hoc analysis, for the primary outcome, we tested the interaction between the randomized intervention effect, and whether the diagnosis based on coronary angiography differed from the diagnosis based on noninvasive endotyping. The interaction p-value is reported, along with the within-subgroup intervention effect estimates.

Secondary outcomes

Secondary outcomes in the randomized trial were analysed in the same way as the primary outcome. Residual distributions were examined visually, and standard transformations were applied where necessary to improve model fit. Comparisons of other secondary and exploratory outcome variables were done using Fisher tests (categorical outcomes) or the Mann-Whitney U test (continuous outcomes) where appropriate.

The Fisher's exact test has been used for the analysis of noncontinuous outcome data in Table 3. This is mentioned in the footnote to Table 3:

"P-values are from the Fisher's Exact test or the Mann-Whitney U test for continuous variables."

27 531 It is quite unusual that the comparative result of the primary outcome is not mentioned in this section, but only in the abstract.

RESPONSE: This point is very well taken. Thank you for highlighting. There is a word count restriction for the abstract. Nonetheless, we have added the following text:

Abstract, page 3, lines 29 – 41

The initial diagnosis based on the angiogram was reclassified by myocardial blood flow imaging in 131 (53.0%, 95% confidence interval: 46.6%, 59.3%) patients.

44 --- The listed percentage in the Intervention arm with “Diagnosis changed from baseline to microvascular angina” is incorrectly stated at 98.4%. Also, it seems misleading to have “Missed diagnosis” in the table. How can we call anything as missed diagnosis without knowing the truth. If “truth” is considered the CMR result, then the study is even more clearly un-ethical!

RESPONSE: Thank you for this point which is well taken. We have changed the term ‘Missed Diagnosis’ to ‘Diagnosis reclassified’.

The percentage 98.4% is correct since it represents 61/62 - i.e. of those with a final diagnosis MVA, 98.4% had changed from baseline

Please note lower down the table - for non-cardiac chest pain, 2/60 (3.3%) had been reclassified; same for “other”.

Reviewer #2 (Remarks to the Author):

Key results

Patients diagnosed with noncardiac chest pain at angiography often have microvascular disease demonstrated on CMR imaging. Management of microvascular disease improves quality of life and may improve long-term prognosis.

Validity

The trial was well conducted, complete with an elaborate blinding scheme. Management was protocol driven to reduce the introduction of bias. Measurement instruments were objective and appropriate. Sample size and statistical analysis were appropriate.

Before changing practice, external validity and reproducibility are needed. Additionally, it is unclear how these findings, if reproduced, would be incorporated into management protocols. This will require future work to build on these findings.

Overall I have no critical or major concerns.

RESPONSE: We are most grateful to Reviewer #2 for their favourable assessment of our study.

Minor concerns and items are listed below.

Significance

These findings are highly significant. Microvascular disease as a cause of chest pain is not novel, nor is diagnosing it via CMR. However, demonstrating that intervening on these patients in a standardized way improves health, in patients who otherwise would have been untreated, has potentially large implications on management of acute chest pain syndromes. This could impact long term survival, healthcare spend, and healthcare utilization.

RESPONSE: Thank you for identifying the importance of this clinical problem and historical under-recognition.

Data and methodology

The data provided are comprehensive, have adequate methods describing their capture, the appropriate tools were used for quality of life measurement. The specific measurements for myocardial blood flow are outside my scope of expertise.

There are some areas that need clarification to ensure this reviewer correctly interpreted the methods, and to ensure readers understand:

1. The distinction between the Radiology report, final diagnosis in the radiology report, and final clinical diagnosis are extremely important and often confusing as written. (lines 227-231 specifically, but also throughout)

RESPONSE: This point is well taken. We have scrutinised the language and made the following changes. The clarifications have been highlighted in yellow in the main text and the supplement.

Page 13, lines 229

The Radiology Report **was formatted and communicated into the medical record** for all participants as would normally be done in standard care. The standard findings that would normally be included in a Radiology Report included cardiac dimensions and function, myocardial tissue characterization, late gadolinium enhancement (absent, present and pattern) and any clinically significant incidental findings (all patients, both groups) were described in the Radiology Report.

The Radiology Report did not include information on the randomized group or the measurements of myocardial blood flow. In the intervention group, the final diagnosis took account of the myocardial blood flow findings, but the actual measurements were not reported. In the control group, the final diagnosis and related treatment were guided by the angiogram but not by the myocardial blood flow findings.

The Radiology Report also included a description of any prognostically important finding that was revealed by CMR imaging e.g. aortic stenosis. Therefore, prognostically important incidental findings were disclosed to the attending clinicians and the participants, as would normally be done in standard care.

Appendix 4 of the protocol provided an SOP for Cardiovascular MRI, including definitions of incidental findings and their management.

2. Lines 200-235 – Please clearly state what was available to form the basis of treatment in each group. All of the information is in these lines, but it required reading numerous times to understand what aspects of information were available in each group. There may actually be too much detail provided.

RESPONSE: Thank you for this feedback. We accept that in attempting to comprehensively convey implementation, the methods text was unduly long. Much of this text has now been moved to the Supplement. Specifically, the text on Medical Management, CMR Acquisition, CMR Analysis, Sample Size Calculation and Data and code availability' have been moved to the Supplement.

Text on "Invasive and Noninvasive Assessment of Diagnosis" (pages 12 – 13), "Randomization, Groups and Blinding" (page 14), and 'myocardial blood flow mapping' (page 15, lines 279 - 286) have been retained since these sections are fundamental to the design of the study.

The word count for the Methods has been reduced from 3,911 to 2,848.

3. Please provide details linking the treatment protocols to any established guidelines.

RESPONSE: Thank you for this feedback.

The CorCMR trial was designed in 2019/2020 and we referred to the guidelines for chronic coronary syndromes published by the European Society of Cardiology (2019). For the first time, these guidelines included dedicated sections of text on microvascular angina.

For implementation, we provided two guidance documents for individual participants. The Clinician Guidance letter provided information on medical management linked to the endotype. This document also included reference to the ESC CCS guidelines (2019). The Patient Guidance letter provided information for their endotype with Plain English language used.

These documents were included in the protocol (Appendix 6 – Clinician Guidance documents; Appendix 7 – Patient Guidance documents). These documents have now been included in the Supplement.

Pages 24 – 28 – Clinician guidance

Pages 28 – 40 – Patient guidance

1. Knuuti J, Wijns W, Saraste A, Capodanno D, Barbato E, Funck-Brentano C, Prescott E, Storey RF, Deaton C, Cuisset T, Agewall S, Dickstein K, Edvardsen T, Escaned J, Gersh BJ, Svitil P, Gilard M, Hasdai D, Hatala R, Mahfoud F, Masip J, Muneretto C, Valgimigli M, Achenbach S, Bax JJ; ESC Scientific Document Group. 2019 ESC Guidelines for the diagnosis and management of chronic coronary syndromes. Eur Heart J. 2019 Aug 31. pii: ehz425.

Analytical approach

I am not a statistical expert. However, I have no concerns with the analysis approach from a clinician scientist perspective.

RESPONSE: Thank you for this favourable feedback.

Suggested improvements

Major concerns: None

Minor items:

RESPONSE: We are greatly encouraged that the Reviewer finds that only minor improvements are indicated

It would be helpful to understand whether the improvements demonstrated in the intervention group were attributable to improvements in patients with microvascular disease or evenly spread across all participants in the intervention group. That would help readers determine whether the pharmacological interventions drove the improvement, or some other unknown variable led to the observed changes.

RESPONSE: We thank the reviewer for this insightful comment. As a result, we have undertaken a subgroup analysis in relation to whether or not the post-CMR diagnosis differed from the initial diagnosis.

We found a highly significant interaction ($p < 0.001$) with no effect of the intervention in those where the post-CMR diagnosis was the same as the angiography-guided diagnosis (intervention effect estimates at 6 months: 3.43 (-2.88, 9.75), $p = 0.285$; at 12 months: 4.12 (-2.67, 10.91), $p = 0.233$), but a large effect of disclosure of CMR results when these led to a different diagnosis (at 6 months: 27.48 (21.44, 33.52), $p < 0.001$); at 12 months: 36.99 (30.49, 43.49), $p < 0.001$). This is in some ways obvious, but worth spelling out, since it shows that the impact of disclosure at the population level is attributable to a large and direct impact in those patients where the CMR result affects the diagnosis.

We have added a section to the paper to include these important findings.

Results, page 24, lines 468 – 475

In a post-hoc analysis following peer-review, we found that the effect of the intervention (disclosure of noninvasive endotyping results) was larger for those patients where noninvasive endotyping led to a different diagnosis compared to angiography (Supplementary Table S3). In those with a different diagnosis, disclosure resulted in an adjusted mean difference in SAQ SS of 27.48 (95% CI: 21.44, 33.52), $p < 0.001$, whereas in those with no change in diagnosis, there was no evidence that disclosure affected SAQ SS (adjusted mean difference 3.43 (-2.88, 9.75), $p = 0.285$); $p_{\text{interaction}} < 0.001$. The magnitude of this effect were even greater at twelve months (Table S3).

Please clarify whether most patients were being referred already for CMR after angiography, or were these patients proactively recruited who otherwise would not have undergone CMR.

RESPONSE: Stress CMR imaging is not approved by National Health Service (NHS) NHS Scotland and is therefore not undertaken in any hospital in Scotland. The NHS is the main provider of healthcare in Scotland.

In the United Kingdom, healthcare is a devolved matter, therefore, the provision of CMR imaging differs markedly between England, Scotland, Wales and Northern Ireland. Stress CMR imaging is widely undertaken in NHS England (especially in London), whereas stress CMR imaging is not undertaken for clinical purposes in NHS Scotland.

In our centre in Glasgow, stress CMR imaging may be undertaken in the NHS if funded through a research grant, as is the case for our current study (BHF Project Grant reference PG/19/28/34310).

Therefore, as indicated by the reviewer, the patients who gave informed consent to participate in this study would not ordinarily have had an MRI scan. This point has been made clear in the Methods.

We have added the following text

Page 9, line 138 – 140

Since stress perfusion cardiovascular magnetic resonance (CMR) imaging is not approved for clinical use in NHS Scotland, the patients who were invited to participate would not otherwise have undergone CMR imaging.

The sentence from 620-622 is not correctly structured.

RESPONSE. Thank you. This sentence is now revised to read as follows:

Page 27 – 28, lines 549 – 550

Overall, the results highlight the diagnostic gap for microvascular angina when coronary angiography is undertaken without adjunctive functional tests.

Clarity and context

Overall the clarity could be improved. There are areas with excessive detail that obscure

the more important big picture of the trial. Such details could be moved to the supplement. Examples: randomization scheme in lines 249-256 could be shortened; lines 306-332 could be grouped conceptually with lines 200-235 and the entire section shortened and clarified.

RESPONSE. Thank you. We have followed the Reviewer's recommendation and this information is now only provided in the Supplement.

Specifically, in the manuscript, these sections of text have been reduced with the statement “**Further details are provided in the Supplement.**”

The section on Definition on Endotypes has now been moved (page 11, lines 215) to precede the Diagnosis section (page 14), and the text re COVADIS criteria has been reduced with guidance to refer to the Supplement.

Lines 428-430 are a high level important message about trial design that need to be up-front and emphasized.

RESPONSE. Thank you. We have emphasized the trial design in the Abstract.

Page 3, Methods, lines 32 – 33

This study was a prospective, multicenter, parallel group, 1:1 randomized, controlled superiority trial.

References

Appropriate

Your expertise

I am a clinician scientist familiar with CMR clinical trials and the evaluation of patients with acute chest pain, most comfortable with clinical trial design and conduct. I am not an imaging expert nor an expert in statistical analysis.

Reviewer #3 (Remarks to the Author):

The authors compared the final diagnosis of patients with angina and no obstructive coronary arteries (ANOCA) on invasive angiography after cardiac MR (CMR). They classified endotypes into microvascular angina, vasospastic angina, obstructive CAD, noncardiac chest pain, and other. Coronary microvascular dysfunction was defined by reduced global myocardial perfusion reserve (MPR), the ratio of stress to rest myocardial blood flow. A threshold of global stress myocardial blood flow $<2.25\text{ml}/\text{min}/\text{g}$, $\text{MPR}<2.2$, or $\text{MPRENDO}<2.41$ was deemed consistent with that diagnosis. They randomized 250 participants with angina and no obstructive disease on invasive angiography who underwent CMR to disclosure of CMR results or non-disclosure. The primary outcome was the reclassification rate of the initial diagnosis (based on coronary angiography) versus the final diagnosis (after non-invasive endotyping). The Seattle Angina Questionnaire (SAQ) summary score and the 5-level EQ-5D questionnaire measured at 6- and 12- months were secondary outcomes. After invasive angiography but before CMR 98% of participants were classified as having non-cardiac chest pain. In the intervention arm, with disclosure of myocardial blood flow findings, 62 (50.0%) had a diagnosis of microvascular angina, a difference of 49.2% (95% CI: 41.3%, 58.7%). A diagnosis of non-cardiac chest pain was predominant in the control arm (98.4%) but much reduced in the intervention arm (48.4%). At six months, SAQ Summary Scores in the intervention and control groups were 67 (18-point improvement from baseline) vs. 54 (0.7-point improvement from baseline). Similar large differences between groups were evident at 12 months.

This is an important study highlighting how frequently we may miss the diagnosis of microvascular angina in patients with ANOCA. It is well designed and the results are compelling and of great clinical relevance.

RESPONSE: We are most grateful to the Reviewer for their favourable assessment of our study.

1. Please mention that although the trial participants had no obstructive disease, they may have had prior MI and PCI.

RESPONSE: Thank you. This point is very well taken. We have revised the manuscript accordingly:

Page 21, lines 406 - 408

One hundred and five (42.0%) participants a history of hospitalization for chest pain, 56 (22.4%) participants had previously undergone coronary angiography, 41 (16.4%) had a history of previous percutaneous coronary intervention and 30 (12.0%) had a history of prior myocardial infarction.

2. Why did you not include myocardial bridges as one of the ANOCA endotypes? Were they included in other?

RESPONSE: Thank you for highlighting. This question is very well taken.

Myocardial bridges are not an endotype, they are an anatomical variant that predisposes to coronary vascular dysfunction and related anginal chest pain.

Myocardial bridges represent the overlying myocardium for a coronary artery with a limited intramuscular passage. This segment of coronary artery is associated with vasomotor dysfunction localised to the segment of coronary artery with the intramural course. Typically, in humans, the mid-segment of the left anterior descending coronary artery is affected.

In a symptomatic patient the presence of angina + myocardial bridge (by angiography) + coronary microvascular dysfunction (functional test) is a diagnostic triad which are likely to be causally associated.

This being said, myocardial bridges commonly occur in asymptomatic individuals. One review on this topic cites up to 30% of the population may have a coronary artery with overlying myocardium (bridge) [PMID: 34136540], and myocardial bridging is also a common incidental finding on CT coronary angiography [PMID: 35969186].

3. Although your study recruited from the cath lab, please speculate in the Discussion as to how this approach to anatomical and functional testing to diagnose coronary microvascular dysfunction might be accomplished with CCTA as the anatomical imaging modality.

RESPONSE: Thank you. Interestingly, in the landmark SCOT-HEART trial (including >800 participants from Glasgow), in the CCTA guided group, angina and quality of life was worse relative to standard care. We have now added the following text to the Discussion:-

Page 29, lines 580 – 582

In the Scottish Computed Tomography of the Heart (SCOT-HEART) trial³¹, compared to standard care based on functional testing, an anatomical strategy using CCTA added to standard care was associated with relatively worse angina symptoms and health-related quality of life³².

References

31. Newby, D. *et al.* CT coronary angiography in patients with suspected angina due to coronary heart disease (SCOT-HEART): An open-label, parallel-group, multicentre trial. *The Lancet* **385**, 2383–2391 (2015).

32. Williams, M. C. *et al.* Symptoms and quality of life in patients with suspected angina undergoing CT coronary angiography: a randomised controlled trial. *Heart* **103**, 995–1001 (2017).

4. The differences in medical therapy are substantial, but it is not entirely clear how this was accomplished so effectively. Please provide for the supplement the standardized letters with customized medical management guidelines that were sent to the general practitioner and cardiologist with advice on treatment optimization to relieve anginal symptoms. Please show a template letter for the intervention group and for the control group (if they are different).

RESPONSE: Thank you.

These documents were included in the protocol (Appendix 6 – Clinician Guidance documents; Appendix 7 – Patient Guidance documents) and the documents have now been included in the Supplement.

Pages 24 – 28 – Clinician guidance

Pages 28 – 40 – Patient guidance

5. Medical therapy for microvascular angina is not established, yet you report impressive improvement in the SAQ scores associated with changes in medical therapy. A comment about this might be considered in the Discussion.

RESPONSE: Thank you. We have now added the following text to the Discussion:-

Page 28, lines 558 – 561

In the current trial, the improvements in angina and health-related quality of life observed over one year in the functional imaging guided group may reflect an improved understanding of medical therapy for microvascular angina and continuity of care post-pandemic.

6. How many participants from each group participated in cardiac rehabilitation?

RESPONSE: Thank you for this question. This information is now included in Table 3 (and highlighted in yellow). Compared to the Control Group, a higher percentage of individuals in the Intervention Group participated in Cardiac Rehabilitation, but the comparison was not statistically significant.

Table 3

	Control N=126	Intervention N=124	p-value
Cardiac rehabilitation, n (%)	35 (27.8%)	48 (38.7%)	p=0.081

7. Please comment on the ability of the average CMR clinical lab to measure myocardial blood flow.

RESPONSE: Thank you. In clinical practice, myocardial perfusion imaging provided by the manufacturer is done by visual assessment or by semiquantitative methods. Fully quantitative methods for performing myocardial perfusion analysis are available from all of the vendors as works-in-progress (WIP) packages which are used with research consent for clinical evaluation. Additionally, several companies offer off-line image analysis software that performs fully quantitative perfusion analysis.

The fully quantitative myocardial blood flow imaging used in our study was developed at NIH and provided as part of a research collaboration. This research software is now available directly from SIEMENS as a works-in-progress (WIP) package. It is anticipated that quantitative myocardial perfusion will be available as product on all vendor platforms in the not too distant future (e.g. 1-2 years).

Minor

1. Preventive is preferred use over preventative in text and tables.

RESPONSE: Thank you. This change has now been made.

2. In Table 1 you wrote “Control” instead of “intervention” in the following: Previous percutaneous coronary Control, n (%).

RESPONSE: Thank you. This term in the statistical report has been corrected.

Reviewer #4 (Remarks to the Author):

This study is a randomized control trial among ANOCA patients. The study showed that with non-invasive endotyping, 53.6% of patients were misdiagnosed based on invasive angiography-guided management. In the intervention group, there were improvements in angina burden and health-related quality of life at 6- and 12-months, as assessed by SAQ and EQ-5D-5L. The authors should be commended for the novelty and clinical relevance of this study, which is supported by a well-designed methodology and a thoughtful discussion. References are appropriate. Appropriate use of statistics. Minor comments are provided below.

RESPONSE: We are most grateful to the Reviewer for their favourable assessment of our study.

1. Study design – Good to see that individuals with lived experience were involved in the protocol development.

RESPONSE: Thank you for highlighting this aspect of the study design.

2. Figure 1 – Further details to this figure are necessary.

RESPONSE: Thank you. We had provided a legend (97-words) for this figure. We have now further expanded the legend (135 words) and the full text is provided below.

In addition, we have now provided two clinical cases with illustrated figures and histories. These figures have been included in the Supplement (Figures S9 – final diagnosis post-CMR, noncardiac chest pain, and Figure S10 – final diagnosis post-CMR, microvascular angina). We have mentioned these cases at the end of the Results

Page 27, lines 528 – 529

Clinical cases

Two illustrated clinical cases are provided in the Supplement (Figures S9 and S10).

Page 36

Figure 1. Study design and blinding status. The attending clinicians with responsibilities for ongoing care and the participants were blind to the randomized group allocation and myocardial blood flow results by CMR imaging. The images in this figure include a left coronary angiogram with no evidence of

atherosclerosis and a color-encoded map of myocardial blood flow (ml/min/g tissue, range 0 – 6.0) during adenosine stress cardiovascular magnetic resonance (CMR) imaging at 1.5 Tesla and myocardial first-pass of gadolinium-based contrast media (0.05 mmol/kg, Gadovist®) given by intravenous bolus. The quantitation of myocardial blood flow is achieved by an in-line automated pixel-wise myocardial perfusion mapping method ²¹. This CMR image reveals a heterogeneous pattern of global myocardial blood flow with a reduction of blood flow in the subendocardium relative to the epicardium. The appearance is consistent with coronary microvascular dysfunction.

3. Line 521-530 – It is stated that no participants had a vasospastic angina diagnosis post-angiogram. However, one participant then had a vasospastic angina diagnosis post-CMR. How did this occur? Did this participant undergo acetylcholine testing and then received a vasospastic angina diagnosis? If so, this needs to be made clearer.

RESPONSE: We apologize for the lack of clarity on this point. The explanation is as follows:

This patient developed significant chest pain with ischaemic ECG changes during the CMR scan. Interestingly, the anginal-sounding chest pain occurred at rest before administration of intravenous adenosine. Following reassurance from the radiology staff, the adenosine stress CMR perfusion scan was commenced and the chest pain recurred during IV adenosine infusion. The participant had an abnormal myocardial perfusion scan on stress imaging.

The final diagnosis based on these observations (clinical and radiological) was mixed vasospastic/microvascular angina. The electronic case report form was not designed to include both endotypes, therefore, the dominant endotype was determined as being vasospastic angina since the chest pain had occurred spontaneously. Presumably, the psychological stress of the CMR examination had rendered the patient to become susceptible to angina at rest. The chest pain resolved spontaneously. The revised diagnosis of probable vasospastic angina was provided to the patient, along with the SOP guideline for this endotype, and the patient returned home.

The CMR exam was recorded with an adverse event, but since no other clinical sequelae occurred, no treatment administered, and the patient was not hospitalised, this episode was not recorded as a serious adverse event.

Page 26, lines 517 – 522

One participant, a 61-year old female with a history of type 2 diabetes mellitus and migraines, experienced spontaneous angina during the rest phase of the CMR examination. This was judged to be secondary to

psychological stress associated with undergoing the scan, and consistent with probable vasospastic angina.

Accordingly, treatment with bisoprolol 2.5 mg daily was changed to a long-acting formulation of diltiazem,

200 mg once daily.

4. Line 616 – is the term ‘vasomotor angina’ referring to microvascular angina? Or is it being used as a combined term for microvascular angina and vasospastic angina?

RESPONSE: We apologize for the lack of clarity on this point. The term vasomotor angina includes either microvascular angina, vasospastic angina or both. As per the variable name in Table 3, we have now deleted ‘vasomotor angina’ in the Discussion and replaced this term with ‘microvascular angina or vasospastic angina.’ This change has been made (page 27, line 545).

5. Line 665 – It is stated that half of the participants in this trial are female which underlines that women are more commonly affected by ANOCA. Is this in reference to coronary artery blockages? This is an unclear statement.

RESPONSE: We apologize for the lack of clarity on this point. We have revised the text and removed the word ‘more’, hence the text now reads as follows:

Page 35, line 717

Finally, half of the participants in this trial were female, underlining that females are commonly affected by ANOCA, with implications for quality of life and morbidity.

6. Line 393 – It is stated that hospital visits for chest pain episodes were documented as a prioritized outcome of interest. The reader may interpret this as an outcome measure, which is not reported in the results.

RESPONSE: Thank you for highlighting this point which is well taken. The variable “Cardiac adverse event: hospital attendance for chest pain, n (%)” is reported in Table 3, penultimate.

Nonetheless, we agree with the Reviewer that this event was not a secondary outcome, therefore we have removed the phrase “were documented as a prioritized outcome of interest” (Methods).

7. Limitation section – It is important to note that with the lack of coronary function testing in this study, this may have resulted in mis-diagnoses of vasospastic angina. Therefore, the post-angiogram post-randomization, post-CMR endotype diagnosis of ‘non-cardiac chest pain’ may include missed diagnoses of vasospastic angina.

RESPONSE: Thank you. This point is well taken. We have now added the following statement to the Limitations:

Page 30, line 609 – 611

Since intracoronary acetylcholine testing was not included, the post-angiogram post-randomization, post-CMR endotype diagnosis of ‘non-cardiac chest pain’ may have include missed diagnoses of vasospastic angina.

We have also added (page 29, lines 580 - 582)

The findings also indicate patient benefits may be achieved from a functional strategy that does not include intracoronary acetylcholine testing.

In summary, we sincerely thank the Editors and Reviewers for their reviewing our manuscript. We hope the revisions and responses are adequate to support a favourable decision.

NMED-FT144896A Response to reviewers
Reviewer #1:

Remarks to the Author:

**Endotyping Patients with Chest Pain and No Obstructive Coronary Artery Disease:
A Randomized Controlled Trial**

The authors have addressed all of my primary concerns.

I do have some concerns still about Table 3. Last time I questioned the reported percentage for “Diagnosis changed from baseline to microvascular angina” which occurred for 61 out of 124 intervention participants while the reported percentage was 98.4. It seems for some rows in Table 2 you give percentages of a subset of participants, in this case the subset of participants with final diagnosis of microvascular angina. That is confusing and not reflected in the left-hand column describing the row. If you really want to give such percentages of a subset (not recommended), then you need to modify the row description accordingly (for example “Diagnosis of those with final diagnosis of microvascular angina was changed from baseline”).

Moreover, the numbers in Table 3 don’t seem to make any sense to me. Perhaps I don’t understand what is meant by reclassified. I see several rows that give numbers of “Diagnosis changed from baseline to X”, which seem to represent disjoint subsets. But when I count up those in the Control and Intervention arms, I get 0 and 65 respectively. Surely this can’t be right. Moreover, I then see “Diagnosis reclassified” of 67 and 0 respectively! Nowhere in this table do I see a total (control+intervention) of (the reported) 131 for the primary cohort outcome of reclassification.

RESPONSE: Thank you for this comment. We have followed the Reviewer’s advice. Table 3 is simplified.

Percentages are removed from individual cells and provided only for the summary cells at the lower row for each group (Control, Intervention).

For clarity the row for ‘Final working diagnosis’ is highlighted in bold type. The summation and percentages for the cells in these rows should now be very clear. The number ‘131’ is no longer mentioned.

The table is provided below.

The p-value for the primary analysis of the diagnostic study is provided in table 2. Therefore, we have not duplicated the same test in Tables 2 and 3.

Table 3. Differences between original diagnosis (based on clinical history and invasive coronary angiogram) and final diagnosis based on additional information following cardiovascular magnetic resonance imaging. The 'final working diagnosis' (bold) was used to guide treatment decisions after randomization.

		Post-cardiovascular magnetic resonance imaging diagnosis					
		Obstructive CAD	Microvascular angina	Vasospastic angina	Non-cardiac chest pain	Other diagnosis	Final working diagnosis (control group)
Control N= 126	Original diagnosis	Obstructive CAD	0	1	0	0	1 (0.8%)
		Microvascular angina	0	1	0	0	1 (0.8%)
		Vasospastic angina	0	0	0	0	0 (0.0%)
		Non-cardiac chest pain	0	63	1	57	124 (98.4%)
		Other	0	0	0	0	0 (0.0%)
		0 (0.0%)	65 (51.6%)	1 (0.8%)	57 (45.2%)	2 (1.6%)	
Intervention N= 124	Original diagnosis	Obstructive CAD	0	1	0	0	1 (0.8%)
		Microvascular angina	0	1	0	2	3 (2.4%)
		Vasospastic angina	0	0	0	0	0 (0.0%)
		Non-cardiac chest pain	0	60	0	58	120 (96.8%)
		Other	0	0	0	0	0 (0.0%)
	Final working diagnosis (intervention group)		0 (0.0%)	62 (50.0%)	0 (0.0%)	60 (48.4%)	2 (1.6%)

CAD – coronary artery disease.

Reviewer #2:

Remarks to the Author:

The authors have completed a detailed and thorough response to the reviewers' concerns. Their responses are adequate and acceptable to this reviewer. The final manuscript is of high quality and significantly advances the medical literature. This reviewer suggests the manuscript be accepted for publication.

RESPONSE: Thank you for this favourable feedback.

Reviewer #3:

Remarks to the Author:

Thank you for your responses to Editor and Reviewer comments. I have no further questions or suggestions.

RESPONSE: Thank you for this favourable feedback.

Reviewer #4:

Remarks to the Author:

Thank you for the opportunity to review this manuscript. I am satisfied with the revisions and comments undertaken in response to the reviewer's comments.

RESPONSE: Thank you for this favourable feedback.